# Learning-Guided Integration Contours Construction for Fast Large-Scale Generalized Eigensolvers

Yeqiu Chen [* 1]   Ziyan Liu [* 1]   Hong Wang [1 †]   Lei Liu [1]

## Abstract

Solving large-scale Generalized Eigenvalue Problems (GEPs) is a fundamental yet computationally prohibitive task in science and engineering. As a promising direction, contour integral (CI) methods offer an efficient and parallelizable framework. However, their performance is critically dependent on the selection of *integration contours*—improper selection without reliable prior knowledge of eigenvalue distribution can incur significant computational overhead and compromise numerical accuracy. To address this challenge, we propose **Deepcontour**, a novel hybrid framework that integrates a deep learning-based spectral predictor with Kernel Density Estimation (KDE) for principled contour design. Specifically, Deepcontour utilizes its specialized Eigen-Neural-Operator (ENO) to provide rapid spectral distribution priors, driving a KDE module to automatically construct the optimized integration contours, which guide the CI solver to efficiently find the desired eigenvalues. Deepcontour achieves up to a 5.63x speedup across diverse scientific datasets while maintaining strict numerical rigor. By merging the predictive power of deep learning with the numerical rigor of classical solvers, this work establishes an efficient and robust paradigm for solving large-scale GEPs.

## 1. Introduction

Generalized eigenvalue problems (GEPs) arise broadly across science and engineering, including structural mechanics (Bathe, 2006), molecular dynamics (Mitsutake & Takano, 2018), quantum chemistry (Szabo & Ostlund, 2012), and stability analysis of dynamical systems (Meirovitch,

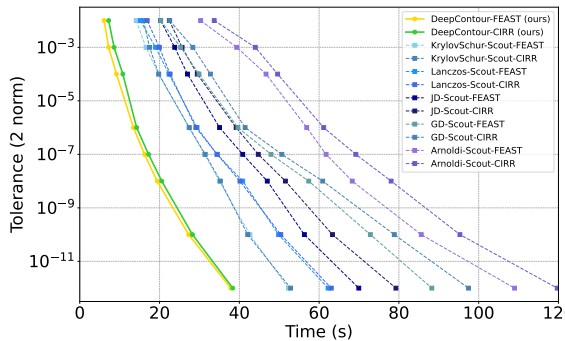

**Figure 1.** The variation in tolerance for Deepcontour compared to scouting methods. Each line shows experimental results using a contour selection strategy and a CI-based solver. Notably, Deepcontour substantially enhances the efficiency of solving GEPs, achieving a speed-up of up to 5.63 times. A comparison with traditional iterative eigensolvers is provided in Appendix G.1. Their significantly slower solving performance also highlights motivation for accelerating CI methods.

2010). They commonly emerge when complex physical or engineered systems are modeled and then discretized into algebraic form, resulting in GEPs whose eigenvalues capture key system behaviors such as resonance and stability (Bathe, 2006; Meirovitch, 2010). For example, in structural vibration analysis, the discretized free-vibration model is written as $K\phi = \omega^2 M\phi$, where the generalized eigenvalues correspond to squared natural frequencies and the eigenvectors describe vibration modes. Specifically, analyzing these characteristics usually requires finding only the eigenvalues located within a specific region of interest. However, as matrix dimensions reach millions or beyond, efficiently finding these eigenvalues within the region remains a significant computational challenge (Saad, 2011; Kressner, 2005).

For large-scale GEPs, *Krylov-subspace methods* (e.g., Lanczos or Arnoldi) have long been the traditional numerical workhorse, demonstrating exceptional efficiency in extracting extremal eigenvalues (Wang et al., 2024; Dong et al., 2024; Wang et al., 2025a; Lehoucq et al., 1998; Saad, 2011). However, these methods encounter severe structural bottlenecks when tasked with finding a large number of interior eigenvalues in high-dimensional systems. Specifically, the

---

[*]Equal contribution   [1]University of Science and Technology of China.   Correspondence to:   Hong Wang <wanghong1700@mail.ustc.edu.cn>.

*Proceedings of the 43rd International Conference on Machine Learning*, Seoul, South Korea. PMLR 306, 2026. Copyright 2026 by the author(s).

shift-and-invert transformation required for interior problems is inherently sequential, which precludes the use of multi-level concurrency (Williams-Young et al., 2020; Aktulga et al., 2014; Wang et al., 2026b). Furthermore, as the matrix dimension and the number of desired eigenvalues increase, the quadratic growth of orthogonalization costs (Nakatsukasa & Tropp, 2024; Kestyn et al., 2016) and the communication overhead from frequent global synchronizations (Ballard et al., 2014; Hoemmen, 2010; Wang et al., 2026a) become prohibitive. To overcome these limitations, **Contour Integral (CI) methods** (Sakurai & Tadano, 2007; Polizzi, 2009; Sakurai & Sugiura, 2003) have emerged as a powerful paradigm, which partition the spectral region of interest (i.e., a prescribed region contains the target eigenvalues) into one or more contours $\{\Gamma_k\}$. For each contour $\Gamma_k$, a corresponding spectral projector, $P_k = \frac{1}{2\pi i} \oint_{\Gamma_k} (zB - A)^{-1} B dz$, is constructed, which isolates the invariant subspace associated with the eigenvalues purely inside that sub-region. This enables solving the original large GEP via smaller, independent subproblems that are naturally parallelizable across contours.

Despite advantages for parallelism, CI methods are critically sensitive to contour construction (Polizzi, 2009; Güttel & Tisseur, 2017). Since spectral projectors are approximated via numerical quadrature along each contour, the dominant computational cost involves solving a sequence of shifted linear systems at each quadrature node (Sakurai & Sugiura, 2003; Beyn, 2012). Therefore, the placement of a contour directly influences both the number of quadrature nodes required to reach a target accuracy and the numerical conditioning of the projected subproblems (Guttel et al., 2015; Gavin & Polizzi, 2016). Improper contour selection severely degrades performance: oversized contours inflate costs due to excessive quadrature nodes and larger projected subproblems (Senzaki et al., 2010), while undersized ones risk missing target eigenvalues (Polizzi, 2009). To ensure efficiency and stability, an ideal contouring strategy must encompass all target eigenvalues while partitioning dense spectral clusters into manageable groups, maintaining a safe distance from the spectrum to avoid quadrature errors (Senzaki et al., 2010; Di Napoli et al., 2016).

In practice, achieving such an optimal strategy is remarkably difficult because the eigenvalue distribution is typically unknown. This lack of prior knowledge forces practitioners to rely on manual heuristics or scouting procedures (Sakurai et al., 2013; Krämer et al., 2013). Manual heuristics are often arbitrary and prone to the risks of either excessive computational waste or missing target eigenvalues (Polizzi, 2009; Senzaki et al., 2010). Traditional scouting—which typically involves a preliminary search using iterative methods—frequently incurs substantial overhead (Sakurai et al., 2013; Krämer et al., 2013). These pre-processing steps can be as time-consuming as the main eigenvalue computation

itself, undermining the parallel efficiency gains that CI methods are designed to provide (Di Napoli et al., 2016; Gavin & Polizzi, 2016). Thus, contour construction remains a primary practical bottleneck that limits the efficiency and reliability of CI-based eigensolvers (Gavin & Polizzi, 2016).

To address this bottleneck, we propose **Deepcontour**, a hybrid framework that bypasses the computational overhead of conventional scouting by leveraging a deep learning-based spectral predictor to rapidly provide reliable prior knowledge for principled contour design. Our method is a two-stage pipeline that aims to automate and improve contour construction. First, we develop the *Eigen-Neural-Operator (ENO)*, a specialized neural architecture designed to rapidly predict the coarse spectral distribution of GEPs with low inference overhead (Li et al., 2020; Choi et al., 2024). Second, leveraging this predicted distribution, we use Kernel Density Estimation (KDE) (Lin et al., 2016; Senzaki et al., 2010) to automatically identify eigenvalue clusters and adaptively construct tight-fitting contours that minimize quadrature errors. To ensure robustness against prediction uncertainties, we incorporate a conservative safety margin in contour placement and can optionally invoke lightweight validation checks. Our framework transforms contour selection into an automated, data-driven procedure while preserving full accuracy under the same tolerances.

In summary, the primary contributions of this work are as follows:

- To the best of our knowledge, our work is the *first* to apply a data-driven approach to optimize the contour construction process within CI methods for efficiently solving large-scale GEPs.

- We introduce a novel contour strategy that integrates a neural operator with a KDE. To facilitate both numerical accuracy and efficiency, we design a specialized network architecture for spectral prediction and an adaptive KDE-based approach for efficient and effective contour construction.

- Extensive experiments on challenging GEPs from five diverse scientific domains demonstrate that our hybrid framework significantly reduces total computational time (with speedups up to $5.63\times$) compared to state-of-the-art scouting-based methods **without compromising numerical precision**.

## 2. Related Works

### 2.1. CI Eigensolvers

CI eigensolvers leverage spectral projection via a complex integral of the resolvent to find all eigenpairs within a region at once. The foundational Sakurai-Sugiura method (SSM)

computes moments from the resolvent to approximate the target eigenvalues (Sakurai & Sugiura, 2003). Building on this, the Contour Integral Rayleigh-Ritz (CIRR) method enhances numerical stability by integrating a Rayleigh-Ritz procedure (Sakurai & Tadano, 2007). FEAST further reformulates the problem as a subspace iteration with a spectrally filtered projector (Polizzi, 2009). Refinements include Z-Pares that replace numerical quadrature with optimized rational approximations to construct the filter more efficiently (Guttel et al., 2015). In practice, CI performance depends heavily on contour placement. Many implementations adopt scouting to guide contouring: this typically involves running a few iterative steps to generate a rough map of Ritz values (Saad, 2011; Gavin & Polizzi, 2016), or using stochastic trace estimation (Hutchinson, 1989; Di Napoli et al., 2016) to approximate the eigenvalue count in a target interval (Sakurai & Sugiura, 2003; Kostic & Voss, 2013). However, these pre-processing steps frequently incur substantial computational overhead (Sakurai et al., 2013; Krämer et al., 2013).

### 2.2. Neural Operators

*Neural Operators* (NOs) are designed to learn mappings between infinite-dimensional function spaces (see Appendix A.4 for a formal definition) (Kovachki et al., 2023). Pioneering architectures, such as the Fourier Neural Operator (FNO) (Li et al., 2020), DeepONet (Lu et al., 2021) and others (Luo et al., 2024; Gao et al., 2024; Wang et al., 2026c; Huang et al., 2025; Hou et al., 2026; Zhang et al., 2026; Dong et al., 2025), are specifically formulated to approximate the solution operators of parametric PDEs, mapping physical parameters to continuous solution fields. In this work, we adapt this powerful operator learning paradigm to directly predict the spectral distribution from governing physical parameters. Crucially, we leverage the resolution invariance (super-resolution) property of FNO (Li et al., 2020), which allows the model to generalize across different mesh resolutions without retraining. This capability enables efficient, low-cost spectral prediction to guide the solver, bypassing the limitations of fixed-discretization networks.

## 3. Preliminaries

### 3.1. Problem Formulation: From PDE Operators to GEPs

Large-scale GEPs studied in this work originate from operator eigenvalue problems defined by partial differential equations (PDEs). These PDEs are parameterized by continuous functions that describe the underlying physical system, such as material density $\rho(\mathbf{x})$, Young's modulus $E(\mathbf{x})$, or geometry-dependent coefficients. We collectively denote these system descriptors as an input function $a(\mathbf{x})$.

Given $a(\mathbf{x})$ and appropriate boundary conditions, a broad class of physical models can be written as a generalized operator eigenproblem: find $(\lambda, u)$ with $u \neq 0$ such that

$$\mathcal{L}_a\, u = \lambda\, \mathcal{M}_a\, u, \qquad (1)$$

where $\mathcal{L}_a$ is a (typically elliptic) differential operator and $\mathcal{M}_a$ is a positive operator (e.g., mass). A standard variational formulation introduces bilinear forms $A_a(\cdot, \cdot)$ and $B_a(\cdot, \cdot)$ induced by $\mathcal{L}_a$ and $\mathcal{M}_a$, respectively, and seeks $(\lambda, u)$ satisfying

$$A_a(u, \phi) = \lambda\, B_a(u, \phi), \qquad \forall \phi \in \mathcal{V}, \qquad (2)$$

for a suitable Hilbert space $\mathcal{V}$ encoding boundary conditions.

Under a Galerkin discretization on a finite-dimensional subspace $\mathcal{V}_h = \mathrm{span}\{\varphi_i\}_{i=1}^n \subset \mathcal{V}$, representing $u_h = \sum_{i=1}^n v_i \varphi_i$, the weak form (2) reduces to the matrix generalized eigenvalue problem

$$A\mathbf{v} = \lambda B \mathbf{v}, \qquad (3)$$

where $A, B \in \mathbb{C}^{n \times n}$ are large, sparse matrices assembled as $A_{ij} = A_a(\varphi_j, \varphi_i)$ and $B_{ij} = B_a(\varphi_j, \varphi_i)$, $\lambda \in \mathbb{C}$ is an eigenvalue, and $\mathbf{v} \in \mathbb{C}^n$ is the corresponding eigenvector (Saad, 2011). Such PDE-induced GEPs are fundamental in science and engineering, e.g., structural vibrations and quantum mechanical states (Bathe, 2006).

In this work, we focus on problems whose spectrum in the target range is *real*. In particular, we consider *Hermitian* GEPs where $A$ and $B$ are Hermitian and $B \succ 0$, which guarantees real generalized eigenvalues and enables robust contour-integral eigensolvers on real-axis spectral windows.

### 3.2. CI Eigensolvers for GEPs

CI eigensolvers are based on a spectral projector defined by complex contour integration. Given a closed contour $\Gamma_k$ in the complex plane, the spectral projector for the GEP $A\mathbf{v} = \lambda B \mathbf{v}$ is defined as:

$$P_k = \frac{1}{2\pi \mathrm{i}} \oint_{\Gamma_k} (zB - A)^{-1} B\, dz, \qquad (4)$$

where i denotes the imaginary unit, and $k \in \{1, \ldots, K\}$ indexes the contours in a partition of the target spectral range.

For any block of vectors $V \in \mathbb{C}^{n \times m}$. Applying the projector in (4) produces $P_k V \in \mathbb{C}^{n \times m}$, whose columns lie in the invariant subspace spanned by eigenvectors with eigenvalues inside the contour $\Gamma_k$. The original large-scale GEP is then projected onto this subspace to yield a small, dense GEP that can be solved effficiently.

In numerical implementations, the contour integral in (4) is evaluated by numerical quadrature, which reduces applying

$P_k$ to $V$ to solving a set of shifted linear systems $(zB - A)Y = BV$ at quadrature nodes on $\Gamma_k$. More details can be referred to Appendix A.3.

Practical implementations of this principle have led to two main families of algorithms: Moment-Based Methods like CIRR (Sakurai & Tadano, 2007) and Subspace Iteration Methods like FEAST (Polizzi, 2009).

### 3.3. Motivation of Deepcontour: Critical Bottleneck of Contour Selection

Regardless of the specific algorithmic variant, the performance of CI methods is critically dependent on the choice of the integration contour(s) $\Gamma$ (Polizzi, 2009; Güttel & Tisseur, 2017; Gavin & Polizzi, 2016). An improperly chosen contour can fail to enclose the desired eigenvalues or be unnecessarily large, leading to excessive computational cost. To empirically show the critical impact of contour selection, we designed a comprehensive *Knowledge-Aware Random* strategy that randomly generates contours based on known ground-truth eigenvalues. We then evaluated the CIRR solver's performance over 100 random seeds for numerous instances and picked 5 representative instances (I1–I5) for presentation, with full details of our strategy design and instance selection provided in Appendix B. Each bar in Figure2 reports the mean and standard deviation (stdev) of two key metrics: the missed eigenvalue rate (reliability) and the total solving time under $10^{-8}$ tolerance (efficiency). The large variance observed demonstrates that CI solver performance is highly sensitive to the contours.

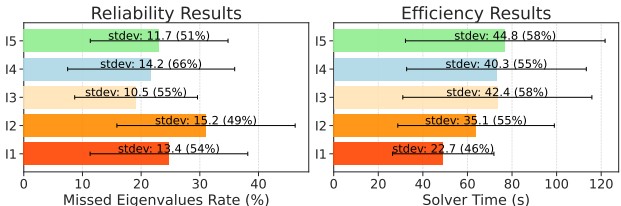

**Figure 2.** Experiments demonstrate the CIRR's high sensitivity to contour selection, as evidenced by the large performance variance observed under a random contour strategy.

However, despite the importance of contour selection, existing strategies have significant limitations. Early automation efforts ranged from manual heuristics to eigenvalue-counting techniques (Sakurai & Sugiura, 2003; Kostic & Voss, 2013; Hutchinson, 1989), but they often lack generality or introduce additional uncertainty. In practice, contour selection involves not only choosing a single $\Gamma$, but also *partitioning* the target range into multiple contours and forming stable projected subspaces (typically via orthogonalization), both of which directly affect end-to-end cost. The current state-of-the-art, *Scouting-based Localization* (Saad, 2011; Gavin & Polizzi, 2016), runs a computationally expensive

iterative solver (e.g., Lanczos) for a fixed number of steps to obtain a coarse spectral map (Ritz values) that guides contour placement. This creates a critical trade-off: map accuracy scales with scouting cost. A cheap scout yields an unreliable map, risking an inefficient or failed solve, whereas an accurate scout undermines the intended cost savings. **Therefore, we propose Deepcontour to address these challenges in contour selection.**

## 4. Method

Deepcontour introduces a hybrid approach that combines the predictive power of deep learning with the numerical rigor of classical solvers to efficiently compute solutions for large-scale GEPs. The method unfolds in two primary stages, as illustrated in Figure 3. First, we employ a custom-designed neural operator, which we term the ENO, to rapidly predict the spectral distribution of a given GEP. Second, we leverage the predicted spectrum to intelligently and automatically construct optimal integration contours for CI methods, thereby overcoming its principal bottleneck. We focus on the real eigenvalues of Hermitian GEPs in this work.

### 4.1. Neural Operator for Rapid Spectral Prediction

The cornerstone of our acceleration strategy is a specialized ENO trained to learn the complex mapping from physical system parameters to their corresponding spectral properties. We provide a detailed discussion in Appendix E about how to build GEP from a PDE problem and a brief introduction to FNO in Appendix A.4.

#### 4.1.1. PROBLEM FORMULATION AS OPERATOR LEARNING

A GEP, as defined in Eq. (3), is fundamentally determined by the underlying physical system. These systems are described by continuous parameter functions, such as material density $\rho(\mathbf{x})$, Young's modulus $E(\mathbf{x})$, or geometric configurations, which we collectively denote as an input function $a(\mathbf{x})$. The $M$ eigenvalues of interest within a prescribed target range form a target vector $\Lambda = (\lambda_1, \ldots, \lambda_M) \in \mathbb{R}^M$. Computing such interior (window) eigenvalues is notoriously difficult for traditional iterative methods (Saad, 2011). Consequently, we can define a solution operator $\mathcal{G}$ that maps the input function $a(\mathbf{x})$ to its first $M$ eigenvalues:

$$\mathcal{G} : a(\mathbf{x}) \mapsto \Lambda. \tag{5}$$

Learning this operator $\mathcal{G}$ allows for the rapid prediction of the spectrum without solving the GEP itself. Since the output $\Lambda$ is a vector of continuous, real-valued numbers, we formulate this task as a multi-output regression problem. Our goal is to train a neural network, the ENO, to approximate this operator $\mathcal{G}_\theta \approx \mathcal{G}$.

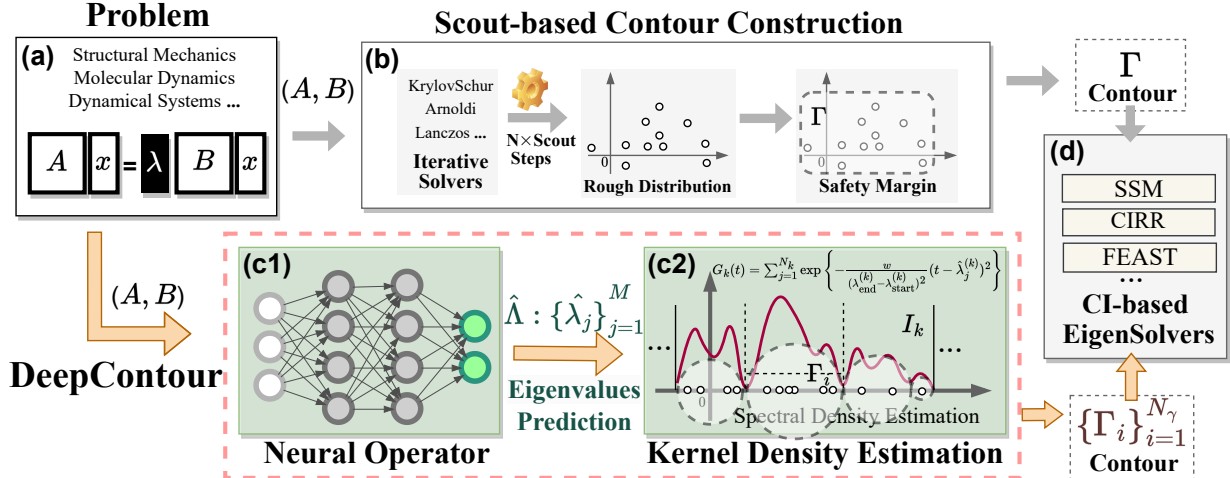

**Figure 3.** Overall architecture of Deepcontour: **(a)** Construct contour $\Gamma$ for CI solver to solve given matrices $A$ and $B$. **(b)** Traditional Scout-based Method: An iterative solver (e.g., Arnoldi) is run for a fixed number of steps to obtain a rough spectral distribution, and a safety margin is then applied to define the integration contour. **(c1) Deepcontour** Eigenvalue Prediction Module: Utilizing a specialized eigenvalue neural operator for estimating target eigenvalues. **(c2) Deepcontour** Adaptive Contour Construction Module: Leveraging KDE to automatically construct tight-fitting, optimized integration contours. (d) Final Solve Stage: The contours is passed to a CI-based Eigensolver (e.g., CIRR) for final solution.

### 4.1.2. THE ENO ARCHITECTURE

The ENO architecture is modularly designed, comprising two main components: (1) FNO-based feature extraction backbone and (2) spectrum prediction head.

**(1) FNO-based Feature Extraction Backbone.** The backbone of our model, denoted as $\mathcal{F}_{\theta_{\text{backbone}}}$, is responsible for processing the input function $a(\mathbf{x})$ and extracting a high-level feature representation that encodes its essential spectral characteristics. We employ the FNO (Li et al., 2020) for this purpose. First, the input function $a(\mathbf{x})$ is discretized on a uniform grid, yielding a tensor $a_{\text{grid}} \in \mathbb{R}^{d_\alpha}$, where $d_\alpha \in \mathbb{N}$. This tensor is lifted by a linear transformation $P$ to a higher-dimensional channel space, creating the initial hidden representation $u_0 = P(a_{\text{grid}})$. This representation $u_0$ is then propagated through a sequence of $L$ Fourier layers, $\{G_l\}_{l=0}^{L-1}$, according to the update rule $u_{l+1} = G_l(u_l)$. Each Fourier layer $G_l$ performs a global convolution in the frequency domain via the Fast Fourier Transform ($\mathcal{F}$), applies a linear transform $R_l$ to the frequency modes, and transforms the result back to the spatial domain with the inverse FFT ($\mathcal{F}^{-1}$), followed by a local transformation $W_l$ and a non-linear activation $\sigma$:

$$u_{l+1}(\mathbf{x}) = \sigma\left(W_l u_l(\mathbf{x}) + \mathcal{F}^{-1}(R_l \cdot (\mathcal{F}u_l))(\mathbf{x})\right). \quad (6)$$

The final hidden representation $u_L$ is then passed through a global pooling operation to produce a fixed-size latent vector $\mathbf{h}$, which serves as the feature-rich summary of the input function: $\mathbf{h} = \text{Pool}(u_L)$.

A practical advantage of using an FNO backbone is its *grid-agnostic* parameterization: its Fourier layers act on a fixed set of modes, so the learned mapping is not tied to a specific grid size. Consequently, once $a(\mathbf{x})$ is resampled onto a uniform grid of resolution $N$ (inducing a GEP of size $n = n(N)$), the trained ENO can be evaluated across different $N$ without retraining.

**(2) Spectrum Prediction Head.** After generating the latent feature vector $\mathbf{h} \in \mathbb{R}^{d_{\theta_{\text{latent}}}}$ from the FNO-based backbone, a prediction head, denoted as $\mathcal{H}_{\theta_{\text{head}}}$, maps this representation to the target output. This head is implemented as a Multi-Layer Perceptron (MLP) (Bengio et al., 2017), $\mathcal{H}_{\theta_{\text{head}}} : \mathbb{R}^{d_{\text{latent}}} \to \mathbb{R}^M$. It projects the high-dimensional features of $\mathbf{h}$ into the desired output dimension $M$, yielding the predicted eigenvalue vector $\hat{\Lambda} \in \mathbb{R}^M$, where each output neuron corresponds to a predicted eigenvalue.

The complete ENO model, $\mathcal{G}_\theta$, is the composition of the backbone and the head, with trainable parameters $\theta = \{\theta_{\text{backbone}}, \theta_{\text{head}}\}$. Let $\mathcal{D} = \{(a_{\text{grid}}^{(i)}, \Lambda^{(i)})\}_{i=1}^{N_s}$ denote a dataset of $N_s$ samples, where $i$ indexes the samples. The model is trained end-to-end by minimizing the Mean Squared Error (MSE) loss between the predicted eigenvalues $\hat{\Lambda}$ and the ground-truth values $\Lambda$:

$$\mathcal{L}(\theta) = \frac{1}{N_s} \sum_{i=1}^{N_s} \left\| \mathcal{G}_\theta(a_{\text{grid}}^{(i)}) - \Lambda^{(i)} \right\|_2^2. \quad (7)$$

### 4.2. Adaptive Contour Design for CI Solver

With a rapid and accurate spectral prediction $\hat{\Lambda}$ from ENO, we can now address the critical challenge of contour selection. Our KDE-based (Lin et al., 2016; Senzaki et al.,

2010) strategy turns the predicted spectrum $\hat{\Lambda}$ into solver-ready CI contours automatically, where KDE smooths the discrete eigenvalue predictions by placing a Gaussian kernel at each predicted eigenvalue to form a simple 1D density estimate. This smoothed profile highlights low-density gaps that naturally separate spectral clusters, and we place contour boundaries at these gaps and wrap each cluster with a tight contour with a small safety margin, yielding robust contours $\{\Gamma_i\}_{i=1}^{N_\gamma}$ for subsequent CI solvers.

**Rationale for KDE-based partitioning.** KDE is well matched to our setting because the Hermitian GEPs considered here have real target spectra, reducing contour construction to a one-dimensional spectral partitioning problem. Direct clustering can be sensitive to local prediction noise and usually requires a pre-specified number of clusters, whereas KDE smooths the predicted eigenvalues into a density profile whose low-density valleys naturally indicate stable split points. This allows contour boundaries to be placed away from dense spectral clusters, reducing quadrature errors and avoiding overly conservative projected subproblems. We do not claim KDE to be globally optimal; it serves as a lightweight and stable partitioning rule for our contour-construction pipeline.

**Interval Sparsity Kernel Function.** A kernel-based sparsity function $G_k(t)$ is introduced to operate on an interval $I_k = [\lambda_{\text{start}}^{(k)}, \lambda_{\text{end}}^{(k)}]$ by summing Gaussian kernels over the predicted eigenvalues in that interval:

$$G_k(t) = \sum_{j=1}^{N_k} \exp\left\{ -\frac{w}{(\lambda_{\text{end}}^{(k)} - \lambda_{\text{start}}^{(k)})^2}(t - \hat{\lambda}_j^{(k)})^2 \right\}. \quad (8)$$

Here, $\{\hat{\lambda}_j^{(k)}\}_{j=1}^{N_k}$ is the subset of $\hat{\Lambda}$ in $I_k$ (re-indexed) and $w$ controls gap sensitivity. The minimizer $t_{\text{cut}}$ marks the sparsest location in $I_k$ and serves as the split point. The resulting sub-intervals are then mapped to contours in the complex plane.

**Rapid Iterative Contour Construction.** Based on this sparsity function, our automated contour construction is a rapid iterative process that enforces a target eigenvalue-count range per contour, namely each contour encloses between $N_{\text{min}}$ and $N_{\text{max}}$ predicted eigenvalues. (1) The process begins by building an initial spectral range

$$I_0 = \left[\min(\hat{\Lambda}) - \epsilon,\ \max(\hat{\Lambda}) + \epsilon\right],$$
$$\epsilon = \beta\left(\max(\hat{\Lambda}) - \min(\hat{\Lambda})\right).$$

where $\epsilon$ is a validation-calibrated global buffer. This buffer expands the predicted spectral hull to compensate for residual prediction error before the KDE-based recursive partitioning is applied. (2) This range is then recursively partitioned by finding the minima of the sparsity function $G_k(t)$,

creating sub-intervals until each contains fewer than $N_{\text{max}}$ eigenvalues. (3) Finally, a refinement loop merges any interval containing fewer than $N_{\text{min}}$ eigenvalues with a neighbor and re-splits any merged interval that violates the $N_{\text{max}}$ limit.

This efficient procedure quickly yields a set of intervals $\{I_k\}_{k=1}^{N_\gamma}$ with well-balanced eigenvalue counts, from which the final, tight-fitting circular contours $\{\Gamma_i\}_{i=1}^{N_\gamma}$ are constructed. For each interval $I_k = [\lambda_{\text{start}}^k, \lambda_{\text{end}}^k]$, the contour $\Gamma_k$ is defined by center

$$c_k = \frac{\lambda_{\text{start}}^k + \lambda_{\text{end}}^k}{2}$$

and radius

$$R_k = (1 + \alpha)\frac{\lambda_{\text{end}}^k - \lambda_{\text{start}}^k}{2}.$$

Here, $\alpha$ is a local inflation factor that keeps the contour boundary separated from the predicted spectral cluster. It is calibrated once on the validation set and then fixed for all test instances. In addition, we use a global buffer

$$\epsilon = \beta\left(\max(\hat{\Lambda}) - \min(\hat{\Lambda})\right),$$

where $\beta$ is a small coefficient selected on the validation set. The initial spectral range is therefore

$$I_0 = \left[\min(\hat{\Lambda}) - \epsilon,\ \max(\hat{\Lambda}) + \epsilon\right].$$

The local factor $\alpha$ controls the compactness of each contour, while the global buffer $\epsilon$ accounts for residual prediction error in the overall predicted spectral hull. In our experiments, both parameters are chosen using only validation instances and are kept unchanged during testing. Following (Senzaki et al., 2010), we employ the $N_q$-point trapezoidal rule with $N_q = 32$ for numerical integration on $\Gamma_k$. Using the minima of $G_k(t)$ as cutting points serves as an **intrinsic safeguard**, as it ensures contour boundaries are positioned at the sparsest regions, thereby minimizing quadrature errors near eigenvalues. A practical advantage of KDE is its low sensitivity to hyperparameters ($N_{\text{min}}$, $N_{\text{max}}$, $w$) (Lin et al., 2016; Senzaki et al., 2010), which were set with minimal tuning in our experiments. The detailed procedures for this flow and complete Deepcontour framework are provided in Appendix (see Alg. 2 and Alg. 3).

# 5. Experimental Results

To comprehensively evaluate the performance of our proposed Deepcontour (ENO-KDE) framework, we conducted a series of extensive numerical experiments. Our evaluation is designed to answer three core questions: (1) How significant is the computational advantage of our framework over traditional scouting-based contour-integral methods?

**Table 1.** Speedup comparison of Deepcontour against five scouting-based baselines. The results are shown for large-scale problems ($N = 50000$) across multiple datasets and for different accuracy tolerances. Each cell presents two metrics in the format: **End-to-End Time Speedup / CI Solver Time Speedup** (refer to Section 5.1 for details). Our method consistently and significantly outperforms all baselines, with all reported speedup values being greater than one.

| Dataset | Tolerance | vs. Arnoldi | vs. GD | vs. JD | vs. Lanczos | vs. KrylovSchur |
|---|---|---|---|---|---|---|
| Kirchhoff-Love Plate | 1e-2 | 5.63 / 4.70 | 4.58 / 3.85 | 4.36 / 3.74 | 2.88 / 2.15 | 2.68 / 2.09 |
| | 1e-4 | 5.25 / 4.27 | 4.42 / 3.68 | 4.31 / 3.23 | 2.71 / 2.04 | 2.55 / 2.01 |
| | 1e-7 | 5.09 / 3.98 | 3.95 / 3.03 | 4.06 / 3.15 | 2.45 / 1.98 | 2.48 / 1.98 |
| | 1e-10 | 4.86 / 3.84 | 3.83 / 2.89 | 3.85 / 2.87 | 2.36 / 1.91 | 2.42 / 1.95 |
| | 1e-12 | 3.45 / 3.48 | 3.86 / 2.81 | 3.73 / 2.95 | 2.26 / 1.87 | 2.39 / 1.87 |
| EGFR Electronic | 1e-2 | 4.87 / 3.24 | 4.15 / 3.12 | 3.81 / 2.85 | 2.43 / 2.10 | 2.32 / 2.02 |
| | 1e-4 | 3.76 / 3.05 | 3.48 / 2.55 | 3.38 / 2.74 | 2.41 / 2.01 | 2.24 / 1.85 |
| | 1e-7 | 3.41 / 2.47 | 3.01 / 2.35 | 2.71 / 2.01 | 2.24 / 1.95 | 2.19 / 1.79 |
| | 1e-10 | 3.25 / 2.26 | 2.78 / 2.21 | 2.79 / 2.05 | 2.01 / 1.88 | 2.12 / 1.70 |
| | 1e-12 | 2.97 / 2.09 | 2.51 / 2.15 | 2.48 / 1.91 | 1.94 / 1.85 | 2.01 / 1.83 |
| EM Cavity | 1e-2 | 3.58 / 2.77 | 3.04 / 2.55 | 3.01 / 2.48 | 2.65 / 2.08 | 2.38 / 2.03 |
| | 1e-4 | 3.43 / 2.64 | 2.98 / 2.48 | 2.91 / 2.41 | 2.44 / 1.99 | 2.29 / 1.94 |
| | 1e-7 | 3.34 / 2.36 | 2.85 / 2.31 | 2.75 / 2.20 | 2.37 / 1.86 | 2.07 / 1.89 |
| | 1e-10 | 3.25 / 2.25 | 2.71 / 2.23 | 2.69 / 2.11 | 2.38 / 1.93 | 2.01 / 1.85 |
| | 1e-12 | 3.03 / 2.36 | 2.66 / 2.28 | 2.61 / 2.14 | 2.04 / 1.84 | 1.96 / 1.72 |
| Piezoelectric Coupled-Field | 1e-2 | 3.39 / 2.84 | 3.08 / 2.58 | 3.01 / 2.45 | 2.41 / 1.89 | 2.21 / 1.97 |
| | 1e-4 | 3.17 / 2.50 | 2.97 / 2.51 | 2.85 / 2.38 | 2.38 / 1.63 | 2.15 / 1.88 |
| | 1e-7 | 3.05 / 2.34 | 2.91 / 2.30 | 2.78 / 2.25 | 2.34 / 1.91 | 2.07 / 1.83 |
| | 1e-10 | 2.98 / 2.22 | 2.75 / 2.21 | 2.71 / 2.14 | 2.21 / 1.85 | 1.96 / 1.75 |
| | 1e-12 | 2.87 / 2.19 | 2.68 / 2.26 | 2.63 / 2.18 | 2.19 / 1.82 | 1.89 / 1.81 |
| Thermal Diffusion | 1e-2 | 3.29 / 2.97 | 3.17 / 2.48 | 2.87 / 2.41 | 2.19 / 1.87 | 2.12 / 1.95 |
| | 1e-4 | 3.14 / 2.51 | 2.88 / 2.39 | 2.74 / 2.35 | 2.11 / 1.80 | 1.98 / 1.86 |
| | 1e-7 | 3.07 / 2.41 | 2.76 / 2.32 | 2.67 / 2.20 | 2.12 / 1.79 | 1.93 / 1.81 |
| | 1e-10 | 2.97 / 2.34 | 2.61 / 2.18 | 2.55 / 2.11 | 2.01 / 1.68 | 1.89 / 1.75 |
| | 1e-12 | 3.02 / 2.42 | 2.46 / 2.21 | 2.39 / 2.15 | 1.93 / 1.78 | 1.81 / 1.74 |

(2) Does this performance advantage generalize to problems of varying scales? (3) Are the key components of our framework—namely, the ENO and KDE modules—critical to its overall performance? This section details our experimental setup, main results, scalability analysis, and ablation results.

### 5.1. Experiment Settings

**Evaluation Perspectives** Our evaluation setting unfolds across three dimensions. (1) *Problem Diversity*: We tested Deepcontour on five GEP datasets originating from different scientific and engineering domains to validate its broad applicability. (2) *Problem Scale*: We also considered five distinct problem scales defined by the matrix dimension $N$. In each case, the number of target smallest (in magnitude) eigenvalues $M$ was set to 1% of matrix dimension $N$.(3) *Solution Accuracy*: We performed solves under eight different tolerance levels (from $10^{-2}$ to $10^{-12}$) to assess the framework's robustness under varying precision requirements.

**Baselines** We compare Deepcontour against the state-of-the-art contour construction strategy, *Scouting-based Localization* (Saad, 2011; Gavin & Polizzi, 2016; Sakurai & Tadano, 2007). It runs a traditional iterative solver (e.g., Arnoldi or Lanczos) for a limited number of steps to obtain a coarse spectral estimate (Ritz values) (Saad, 2011; Gavin & Polizzi, 2016), then constructs the integration contour for the subsequent high-precision solver (CIRR in our

case) (Sakurai & Tadano, 2007). Specifically, it forms a tight bounding box around the target Ritz values and expands it by a safety margin to enclose all corresponding eigenvalues. For fairness, we fix the scout iterations to $k = 60$ (after tuning the cost–accuracy trade-off) and set the safety margin to the minimum that still encloses all target eigenvalues, ensuring optimal efficiency without accuracy loss. Under this protocol, we use five scouting algorithms as baselines: (1) *Arnoldi-Scout* (Saad, 2011); (2) *GD-Scout* (Absil et al., 2008); (3) *JD-Scout* (Sleijpen & Van der Vorst, 2000); (4) *Lanczos-Scout* (Saad, 2011) (symmetric problems); and (5) *KrylovSchur-Scout* (Hernández et al., 2007). We use PETSc (Balay et al., 2019) and SLEPc (Hernandez et al., 2005) (v3.23.4) to implement the CI eigensolve and all scouting algorithms; further details are in Appendix F.2.

**Datasets** We generated datasets from five common physical domains. (1) *Kirchhoff-Love Plate Vibration Analysis*: Simulating the natural frequencies and modes of a thin plate structure, a classic problem in structural mechanics (Leissa, 1969; Reddy, 2006). (2) *EGFR Electronic Structure Calculation*: Computing the electronic structure of Epidermal Growth Factor Receptor (EGFR) molecule, representing a typical class of GEPs in quantum chemistry (Szabo & Ostlund, 2012; Roothaan, 1951). (3) *Electromagnetic Cavity Modal Analysis*: Solving for the eigenmodes of an electromagnetic resonator in the TE mode, widely used in RF

engineering (Jin, 2015). (4) *Piezoelectric Coupled-Field Modal Analysis*: Analyzing the acoustic resonance modes in a 2D enclosed space, a fundamental problem in acoustics design (Tiersten, 1969; Allik & Hughes, 1970). (5) *2D Thermal Diffusion Modal Analysis*: Investigating the stability and growth rates of small perturbations in a 2D fluid system, critical to fluid dynamics (Gurtin, 1982; Ezekoye, 2016; Bathe, 2006). For in-depth exposition of the dataset and its generation, kindly refer to Appendix F.3. **The generated datasets will be released upon the paper's acceptance.**

**Metrics** We assess our framework's performance from two critical perspectives using two distinct metrics: **End-to-End Time Speedup**: This primary metric measures the overall efficiency gain of our entire pipeline, from initial prediction/scouting to the final solution. It is defined as the ratio of the total time taken by the baseline approach to that of our proposed framework:

$$\text{Speedup}_{\text{End-to-End}} = \frac{\text{Time}_{\text{Baseline (Scouting + CI Solve)}}}{\text{Time}_{\text{Ours (Hybrid Contour Design + CI Solve)}}}$$

This metric measures the full practical advantage of our method. **CI Solver Time Speedup**: To specifically quantify the quality of the generated contour itself, the second metric isolates the performance of the CI solver. It compares the time taken by the CI solver using the contour from the baseline against using the contour from our method:

$$\text{Speedup}_{\text{Solver}} = \frac{\text{Time}_{\text{CIRR (with Baseline's Contour)}}}{\text{Time}_{\text{CIRR (with Our Contour)}}}$$

A higher value for this metric directly demonstrates that our framework produces a more effective contour (e.g., covering all eigenvalues with smaller area), which accelerates the final high-precision computation.

All experiments were conducted on a compute node equipped with an Intel Xeon Gold 6246R CPU and an NVIDIA RTX4090 GPU. Deep learning components of our framework are trained and accelerated on the GPU, while the CI solvers and all baseline methods were executed in parallel on the CPU, leveraging all 20 available physical cores via OpenMP. For details of experimental settings and hyperparameters, please refer to Appendix F.

### 5.2. Main Experiment

In this section, we focus on the most challenging, large-scale scenario where the matrix size is $N = 50000$ and number of target smallest eigenvalues is $M = 500$. We compare our Deepcontour framework against five scouting-based baselines using CIRR as the CI solver. The results from a similar comparison against the FEAST solver are detailed in Appendix G.1. We also provide comparsion with traditional iterative solvers in Appendix G.1. We evaluate performance from two key perspectives: the **End-to-End Time Speedup**

and the **CI Solver Time Speedup**, with results presented across a range of accuracy tolerances. Table 1 presents these speedups for each dataset. The results demonstrate that our framework maintains a robust and significant performance advantage. The end-to-end speedup is particularly notable in scenarios with lower accuracy requirements, where the cost of scouting constitutes higher portion of baselines' total runtime. For instance, when solving the Kirchhoff-Love Plate problem, our method achieves a remarkable end-to-end speedup of **5.63x** over the standard Arnoldi-based scout.

Furthermore, the CIRR Solver Time Speedup metric consistently shows significant gains, which confirms the superior quality of our generated contours. By producing a tighter and more precise contour, our method significantly reduces the workload on the subsequent CIRR solver. While the absolute solve times for all methods increase as the tolerance becomes stricter, our methods remains superior performance over baselines. These results confirms that our strategy is not only faster but also a more efficient and scalable approach for high-accuracy scientific computing.

### 5.3. Generalization to Matrix Size

To assess the scalability of our framework, we investigated how its performance advantage evolves with increasing problem size. Figure 4 illustrates the speedup of our method relative to the Arnoldi and GD scouts on two representative problems as the matrix size varies. The results indicate that the acceleration effect of our framework becomes more pronounced as the matrix size increases. This excellent scalability demonstrates that our framework is exceptionally well-suited for tackling the large and challenging GEPs.

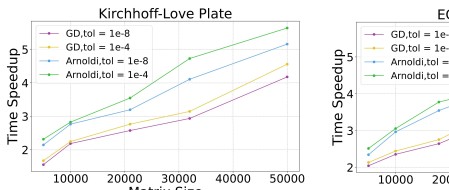

**Figure 4.** Experiments on the *Kirchhoff-Love Plate* and *EGFR Electronic* problems with varying matrix sizes. The results indicate that as the matrix size increases, both time speedup and iteration speedup increase.

### 5.4. Ablation Study

To validate the two core modules in our framework—ENO and KDE—we conducted an ablation study. We compared the following three model configurations: (1) Deepcontour (ENO + KDE): The complete model. (2) w/o ENO: Replace FNOwith a standard MLP, while KDE is retained. (3) w/o KDE: Uses ENO for prediction but replaces KDE with a naive interval expanding process to generate contours. We evaluated these three configurations on one hundred in-

stances from the Kirchhoff-Love Plate dataset ($N = 50000$). As shown in Table 2, Deepcontour achieved the fastest solve time with zero missed eigenvalues. In contrast, the w/o ENO model missed $32.4$ eigenvalues on average. Notably, its KDE contouring module received same tuning effort. The w/o KDE variant increased solve times by over $1.5\times$. This is because its non-adaptive contours must be made conservatively large to guarantee coverage, leading to over-sized and inefficient projection subspaces. To be noted, ENO performs better than scout-based methods even without KDE. This result suggests that KDE mainly improves contour efficiency rather than merely eigenvalue coverage: by adaptively splitting low-density spectral gaps, it avoids unnecessarily large contours and reduces the size of the projected subproblems.

**Table 2.** Ablation study results on the Kirchhoff-Love Plate test case ($N = 50000$). # of Missed $\lambda$ denotes average numbers of uncovered eigenvalues and solve time denotes CIRR solving times under tolerance of 1e-7. Deepcontour performs best in both accuracy and efficiency.

| Model | # of Missed $\lambda$ | Solve Time (s) |
|---|---|---|
| Deepcontour | 0 | 25.2 |
| w/o ENO | 32.4 | 27.4 |
| w/o KDE | 0 | 44.5 |

### 5.5. More Results

For a comprehensive evaluation of our framework, we provide additional results in Appendix G. **(1)** To further demonstrate advantages of our approach, we provide: (i) detailed analysis of predicted spectrum and generated contours (Appendix G.2) and (ii) an ablation study that replaces ENO with traditional scouting methods (Scout+KDE), confirming that accurate spectral prediction is critical for generating high-quality contours (Appendix G.3). **(2)** We further investigate the neural operator component by analyzing its sensitivity to key hyperparameters and comparing the FNO backbone against other neural operators (Appendix G.3). **(3)** We evaluate *super-resolution* of the ENO via a train-low/test-high protocol, demonstrating transfer to higher discretization sizes without finetuning (Appendix G.4). **(4)** We provide detailed runtime breakdown results (Appendix G.5). (5) We further evaluate the robustness of Deepcontour under mild distribution shifts by testing on extrapolated parameter ranges and report both prediction- and solver-level metrics in Appendix G.6. We further analyze how the quality of ENO spectral prediction affects downstream inference speedup and eigenvalue coverage in Appendix G.8.

## 6. Conclusion

We introduce Deepcontour, a novel hybrid framework designed to address the critical contour selection bottleneck in CI methods for solving large-scale GEPs. By synergizing a predictive ENO with a KDE pipeline, our approach transforms the manual, heuristic-based contour construction process into an automated, data-driven strategy. Extensive experiments demonstrate that Deepcontour significantly accelerates GEP solving for CI eigensolvers. For a discussion of limitations and future works, refer to Appendix H.

## Acknowledgment

This research is supported by Smart-Grid National Science and Technology Major Project (Grant No. 2025ZD0805500).

## Impact Statement

This paper advances machine learning methods for accelerating contour-integral eigensolvers by learning to construct robust integration contours for large-scale generalized eigenvalue problems. By reducing the computational overhead and improving the reliability of spectrum-localization pipelines, the proposed approach can lower the cost of scientific and engineering simulations that depend on interior eigenpairs, with potential benefits for applications in structural dynamics, acoustics, electromagnetics, and related domains. The method is intended for numerical analysis and scientific computing; it does not introduce new mechanisms for collecting personal data, nor does it directly enable downstream decision-making about individuals. As with other learning-based numerical components, performance depends on training-data coverage and problem distribution shift; misuse outside the validated regime could lead to incorrect spectral inclusion and degraded solver outcomes. We therefore emphasize transparent reporting of failure modes, conservative safety margins, and validation checks when deploying the method in safety- or mission-critical settings.

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

# A. Extended Related Work and Background

## A.1. Traditional Iterative Algorithms

In computational mathematics, solving the GEPs is a foundational task. While direct methods like the QZ algorithm (Moler & Stewart, 1973) are robust, their $O(n^3)$ complexity is prohibitive for large-scale systems, leading to the dominance of iterative methods for sparse matrices. Krylov subspace methods like the Lanczos and Arnoldi algorithms are highly effective for extremal eigenpairs (Lehoucq et al., 1998; Saad, 2011; Stewart, 2002), while techniques like the Jacobi-Davidson algorithm target specific interior pairs without costly inversions (Sleijpen & Van der Vorst, 2000). However, these solvers become inefficient when numerous eigenvalues within a broad spectral region are required.

## A.2. Neural Network-based Approaches for Eigenvalue Problems

Recent advancements in leveraging neural networks for eigenvalue problems have yielded significant architectural innovations across various scientific domains. Notably, Spectral Inference Networks (SpIN) model eigensolving as a kernel optimization task resolved via deep learning (Pfau et al., 2018). To alleviate the computational burden of the orthogonalization process—a primary bottleneck in high-dimensional spectral analysis—Neural Eigenfunctions (NeuralEF) streamline the pipeline by directly optimizing the projection steps (Deng et al., 2022). Similarly, Neural Singular Value Decomposition (NeuralSVD) integrates truncated SVD for low-rank approximations to enforce the functional orthogonality required for stable representation learning (Ryu et al., 2024). While these end-to-end models offer impressive inference speeds, achieving the high-precision requirements of large-scale scientific applications remains a fundamental challenge.

Another prominent trajectory focuses on the variational optimization of the Rayleigh quotient. The Deep Ritz Method (DRM) employs this principle to compute the smallest eigenvalues, demonstrating substantial potential in solving variational partial differential equations (Yu et al., 2018). This paradigm is frequently extended through Physics-Informed Neural Networks (PINNs) (Ben-Shaul et al., 2023; 2020), which construct variation-free eigenfunctions by embedding governing physical laws into the loss function. Subsequent refinements have introduced specialized regularization terms to enhance the learning accuracy of the lower spectrum (Jin et al., 2022). Beyond variational approaches, some frameworks reformulate the eigenvalue problem as a fixed-point task within semigroup flows, addressed via Diffusion Monte Carlo methods. While iterative strategies such as the Power Method Neural Network (PMNN) effectively approximate individual eigenpairs, the robust and simultaneous extraction of multiple distinct eigenvalues remains a persistent challenge in the field (Yang et al., 2023). A recent advancement is the STNet, which enhances iterative convergence by applying deflation projections and filter transforms directly to the operator (Wang et al., 2025b). While these methods improve iterative solvers, our *Deepcontour* framework adopts a distinct hybrid paradigm, using neural prediction not as a solver but as an intelligent guide for classical algorithms.

## A.3. Numerical Implementation of CI Methods

This section provides further details on the numerical implementation of the CI methods discussed in the main text.

### A.3.1. NUMERICAL QUADRATURE

In practice, the CI in Eq. (4) is computed numerically using a quadrature rule. The contour $\Gamma_k$ is discretized into $N_q$ points $\{z_j\}$ with corresponding weights $\{\omega_j\}$, transforming the integral into a sum:

$$P_k V \approx \sum_{j=1}^{N_q} \omega_j (z_j B - A)^{-1} BV. \qquad (9)$$

The dominant computational cost lies in solving the $N_q$ shifted linear systems of the form $(z_j B - A)Y_j = BV$, which can be done in parallel.

### A.3.2. CI PIPELINE WITH SPECTRAL PARTITIONING

In large-scale settings, a broad target range is often partitioned into multiple sub-intervals, each mapped to a separate contour, so that each projected subproblem remains well-conditioned and small. Algorithm 1 summarizes a standard CI pipeline that combines contour construction via spectral partitioning with orthogonalization and Rayleigh–Ritz projection.

---

**Algorithm 1** Standard CI pipeline with contour partitioning and orthogonalization (high-level)

---

**Require:** Matrices $A$, $B$; target real range $[\lambda_{\min}, \lambda_{\max}]$; partition rule $\mathcal{P}$; quadrature nodes/weights $\{(z_{j,k}, \omega_{j,k})\}_{j=1}^{N_q}$ for each contour; block size $r$
**Ensure:** Eigenpairs in $[\lambda_{\min}, \lambda_{\max}]$
1: Partition the target range: $\{I_k\}_{k=1}^{K} \leftarrow \mathcal{P}([\lambda_{\min}, \lambda_{\max}])$
2: Construct contours: for each $I_k$, build $\Gamma_k$ that encloses $I_k$ with a safety margin
3: **for** $k = 1, \ldots, K$ **in parallel do**
4:     Draw a random block $V \in \mathbb{C}^{n \times r}$
5:     $Y \leftarrow \mathbf{0}$
6:     **for** $j = 1, \ldots, N_q$ **do**
7:         Solve $(z_{j,k}B - A)X_j = BV$
8:         $Y \leftarrow Y + \omega_{j,k}X_j$
9:     **end for**
10:    Orthonormalize $Q \leftarrow \text{orth}(Y)$
11:    Project: $A_k \leftarrow Q^*AQ, \quad B_k \leftarrow Q^*BQ$
12:    Solve the reduced GEP: $A_k u = \lambda B_k u$
13:    Recover Ritz vectors $x \leftarrow Qu$ and keep $\lambda \in I_k$
14: **end for**
15: Merge and de-duplicate solutions across $k$

---

### A.3.3. ALGORITHM FAMILIES

The practical implementation of spectral projection leads to two main families of algorithms: **(1) Moment-Based Methods (e.g., CIRR/SSM)**: This approach, pioneered by Sakurai and Sugiura (Sakurai & Sugiura, 2003), constructs the desired subspace by computing the moments of the resolvent. Instead of computing the projector $P_k$ directly, it computes a sequence of moment matrices for $m = 0, 1, \ldots, M-1$:

$$S_m = \frac{1}{2\pi i} \oint_{\Gamma_k} z^m (zB - A)^{-1} BV \, dz.$$

The subspace is then formed from the span of these moment matrices, $\{S_0, \ldots, S_{M-1}\}$. This technique effectively builds a basis for a Krylov-like subspace in the spectral domain. **(2) Subspace Iteration Methods (e.g., FEAST)**: The FEAST algorithm (Polizzi, 2009) formulates the problem as a subspace iteration, where the spectral projector $P_k$ acts as an ideal filter. Starting with an initial guess subspace $U_0$, the iteration proceeds as $U_{i+1} = \text{orthonormalize}(P_k U_i)$. This process rapidly converges to the target invariant subspace, as applying the projector annihilates vector components corresponding to eigenvalues outside the contour.

### A.4. Brief Introduction to FNO

Neural operators are designed to learn mappings between infinite-dimensional function spaces (Kovachki et al., 2023). We consider an operator $\mathcal{G} : \mathcal{A}(D; \mathbb{R}^{d_a}) \to \mathcal{U}(D; \mathbb{R}^{d_u})$, where $\mathcal{A}$ and $\mathcal{U}$ are Banach spaces of functions defined on a domain $D \subset \mathbb{R}^d$. A neural operator, $\mathcal{N}$, approximates this mapping. For end-to-end training, the continuous functions are discretized into instance pairs $(a, u)$. The purpose is to learn the mapping $\mathcal{N}$ such that $u = \mathcal{N}(a)$.

The mapping $\mathcal{N}$ typically consists of several sequential steps. First, an input channel is lifted to a higher-dimensional representation using a lifting operator $R$. Next, the mapping is performed through a sequence of $L$ iterative layers $\{L_1, L_2, \ldots, L_L\}$. Finally, the output is projected back to the target channel space using a projection operator $Q$. The overall architecture can be expressed as:

$$\mathcal{N}(a) = Q \circ L_L \circ \cdots \circ L_1 \circ R(a). \tag{10}$$

The operators $Q$ and $R$ are pixel-wise transformations and can be implemented using models like an MLP.

The FNO is an effective and widely used architecture for the iterative layers (Li et al., 2020). The innovation of the FNO lies in how it implements the global convolution within each layer. A typical Fourier layer combines a pixel-wise linear transformation (with weight $W$ and bias $b$) with an integral kernel operator $\mathcal{K}$:

$$v_{l+1}(\mathbf{x}) = \sigma \left( W_l v_l(\mathbf{x}) + (\mathcal{K}v_l)(\mathbf{x}) \right), \tag{11}$$

where $v_l$ is the representation from the previous layer and $\sigma$ is a non-linear activation function. The integral kernel operator $\mathcal{K}$ performs the global convolution efficiently by leveraging the Fourier domain. It first transforms the input $v_l$ to the frequency domain using the Fast Fourier Transform (FFT), then applies a linear transformation (a filter) directly to the Fourier modes, and finally transforms the result back to the spatial domain using the inverse FFT. This mechanism allows the FNO to learn global dependencies in a computationally efficient and discretization-invariant manner.

## B. Details of the Validation Experiment of Contour Selection Bottleneck

### B.1. Objective

The primary objective of this experiment is to empirically show the sensitivity of the CI eigen-solver to the geometry of the integration contour. By demonstrating that various contours lead to high performance variance, we establish a clear motivation for an intelligent and robust contour design framework.

### B.2. Problem Instance Selection

We generates 500 problem instances drawn from two scientifically diverse and complex domains—the *Kirchhoff-Love Plate* and *Piezoelectric Coupled-Field* datasets—across a range of matrix sizes (from $N = 5000$ to $N = 25000$ with 1% smallest eigenvalues as target). From these instances, we carefully selected five representative instances, denoted I1 through I5, for detailed presentation. The selection was guided by the principle of covering a wide and varied range of spectral characteristics to demonstrate the broad nature of the contour selection bottleneck. Specifically, the chosen instances represent different scales and spectral complexities. *(1) Instance I1*, Kirchhoff-Love instance with N=10000, was selected for its relatively structured and uniformly spaced spectrum. *(2) Instance I2*, also from the Kirchhoff-Love dataset with N=10000, features the more common scenario of a mix of well-separated low-frequency eigenvalues and more densely clustered higher-frequency modes. *(3) Instance I3*, from the Piezoelectric Coupled-Field dataset with N=10000, was chosen to represent the complexity of coupled-field physics, exhibiting a spectrum with multiple, distinct groups of clusters. *(4) Instance I4*, a Piezoelectric problem with N=25000, is characterized by an extremely dense cluster of eigenvalues within a narrow window. Finally, *(5) Instance I5*, a Kirchhoff-Love problem with N=5000, was selected to represent cases with a large spectral gap between a few dominant modes and the rest. Our analysis is comprehensive and not limited to a single type of eigenvalue distribution.

### B.3. Knowledge-Aware Random Contour Strategy

To ensure our test was both random and meaningful, we designed a *Knowledge-Aware Random* strategy that generates plausible sets of contours based on the ground-truth spectrum of each problem instance. This prevents the generation of trivially poor contours (e.g., those located far from any eigenvalues) and instead simulates a more realistic scenario of uncertainty in both the partitioning and placement of contours. The procedure for generating a single random set of contours for a given instance is as follows:

1. **Determine Number of Contours**: First, we randomly determine the number of contours to generate, $N_\gamma$, from a discrete uniform distribution, $N_\gamma \sim U(\{1, 2, \ldots, 16\})$. This simulates the uncertainty in how many distinct eigenvalue clusters a heuristic method might identify.

2. **Partition Spectral Bounds**: Given the set of ground-truth eigenvalues $\Lambda = \{\lambda_1, \ldots, \lambda_M\}$, we identify the minimal bounding interval on the real axis, $[\lambda_{\min}, \lambda_{\max}]$. We then randomly generate $N_\gamma - 1$ cut points within this interval to partition it into $N_\gamma$ disjoint sub-intervals, $\{I_k = [\lambda_{\text{start}}^{(k)}, \lambda_{\text{end}}^{(k)}]\}_{k=1}^{N_\gamma}$.

3. **Randomly Sample Center and Radius**: For each sub-interval $I_k$, we generate a corresponding contour. A center point $c_k$ is sampled from a uniform distribution over that sub-interval, $c_k \sim U(\lambda_{\text{start}}^{(k)}, \lambda_{\text{end}}^{(k)})$. A radius $r_k$ is then sampled from a log-uniform distribution, $r_k \sim \log U(r_{\min}^{(k)}, r_{\max}^{(k)})$.

4. **Define Radius Range**: The range for the radius sampling is defined based on the spectral properties to ensure plausibility. The lower bound $r_{\min}^{(k)}$ is set to half of the average spectral gap within the sub-interval $I_k$, denoted as $(\lambda_{\max} - \lambda_{\min})/2M$. The upper bound $r_{\max}^{(k)}$ is set to $(\lambda_{\max} - \lambda_{\min})/N^\gamma$.

5. **Construct Contours**: Finally, a set of $N_\gamma$ circular contours $\{\Gamma_k\}_{k=1}^{N_\gamma}$ is constructed in the complex plane, each with its corresponding center $c_k$ and radius $r_k$.

This process was repeated 100 times for each of the five instances, using a different random seed for each run, to generate the full set of contours for evaluation.

### B.4. Evaluation Protocol

For each of the 100 randomly generated contours per instance, we executed the CIRR solver and recorded two key performance metrics:

- **Reliability:** The percentage of ground-truth eigenvalues that were *not* found by the solver within the given contours (Missed Eigenvalues Rate).

- **Efficiency:** The total wall-clock time in seconds for the CIRR solver to converge to a residual tolerance of $10^{-8}$ (Solver Time).

The mean and standard deviation of these metrics over the 100 runs were then calculated and are presented in the main text to illustrate the performance variance.

### B.5. More Results

To further demonstrate the inherent performance sensitivity of CI solvers to the contour's geometry, we conducted a large-scale stochastic evaluation. This experiment simulates the real-world scenario where a practitioner roughly knows the spectral region but must rely on heuristics for the precise placement and size of the integration contour. The experimental setup is as follows: we employed our *Knowledge-Aware Random Strategy*, as detailed in Appendix B, to generate a set of plausible yet varied contours for each problem instance. To show that this performance variability is a general phenomenon, we performed this analysis comprehensively across five of scientific datasets. For each dataset, we randomly selected 100 problem instances at the $N = 50000$ scale. The CIRR solver was then executed on each instance multiple times, each guided by a different randomly generated contour under different random seeds. Table 3 summarizes the aggregated results of this extensive study. It reports the Coefficient of Variation (CV), i.e., $\frac{std}{mean}$, of the average CIRR solve time on 100 instances across 10 random seeds, which measures the performance variability as a percentage.

**Table 3.** Performance variability of the CIRR solver under the Knowledge-Aware Random Contour Strategy. The Coefficient of Variation (CV) is calculated over 100 instances from each dataset. The consistently large CV values highlight the solver's significant sensitivity to contour selection across all domains.

| Dataset | Coeff. of Variation (CV, %) |
|---|---|
| Kirchhoff-Love Plate | 64.1% |
| EGFR Electronic | 58.0% |
| EM Cavity | 61.0% |
| Piezoelectric Coupled-Field | 53.0% |
| Thermal Diffusion | 52.9% |

The coefficient of variation consistently exceeds 50% for most datasets, indicating that the solving time for the baseline method is sensitive to minor heuristic changes in the contour definition. In practice, this means a user can experience dramatically different (and unpredictable) computational costs for the same problem. This high variance underscores the fundamental weakness of relying on heuristics for contour selection and highlights the critical need for a robust approach.

## C. Algorithmic Details

This section provides a detailed, step-by-step description of the key algorithms that constitute our proposed Deepcontour framework. We present two core algorithms. The first, Algorithm 2, details the contour construction process of KDE. The second, Algorithm 3, outlines the complete flow of Deepcontour.

## D. Algorithmic Details

This section provides a detailed, step-by-step description of the key algorithms that constitute our proposed Deepcontour framework. We present two core algorithms. The first, Algorithm 2, details the contour construction process of KDE. The second, Algorithm 3, outlines the complete flow of Deepcontour.

---

**Algorithm 2** Adaptive Contour Construction via KDE

---

1: **Input:** Predicted eigenvalue spectrum $\hat{\Lambda} = \{\hat{\lambda}_j\}_{j=1}^M$; contour parameters $N_{\min}, N_{\max}, w$.
2: **Output:** A set of optimized contours $\{\Gamma_k\}_{k=1}^{N_\gamma}$.
3: Initialize a list of intervals to be processed, $\mathcal{I}_{\text{process}} \leftarrow \{[\min(\hat{\Lambda}), \max(\hat{\Lambda})]\}$.
4: Initialize an empty list for the final contours, $\mathcal{C}_{\text{final}} \leftarrow \emptyset$.
5: **while** $\mathcal{I}_{\text{process}}$ is not empty **do**
6:     Pop an interval $I_k = [\lambda_{\text{start}}^{(k)}, \lambda_{\text{end}}^{(k)}]$ from $\mathcal{I}_{\text{process}}$.
7:     Let $N_k$ be the number of predicted eigenvalues $\{\hat{\lambda}_j^{(k)}\}$ inside $I_k$.
8:     **if** $N_k \leq N_{\max}$ **then**
9:                                             ▷ The interval is valid or too small; construct contour and finalize.
10:         Construct a circular contour $\Gamma_k$ centered at $(\lambda_{\text{start}}^{(k)} + \lambda_{\text{end}}^{(k)})/2$ with radius $(\lambda_{\text{end}}^{(k)} - \lambda_{\text{start}}^{(k)})/2$.
11:         Add $\Gamma_k$ to $\mathcal{C}_{\text{final}}$.
12:     **else**
13:                                         ▷ The interval is too dense; partition it at the sparsest point.
14:         Define the sparsity function for $I_k$: $G_k(t) = \sum_{j=1}^{N_k} \exp\{-\frac{w}{(\lambda_{\text{end}}^{(k)} - \lambda_{\text{start}}^{(k)})^2}(t - \hat{\lambda}_j^{(k)})^2\}$.
15:         Find the point of maximum sparsity: $t_{\text{cut}} \leftarrow \arg\min_{t \in I_k} G_k(t)$.
16:         Split $I_k$ into two new sub-intervals: $[\lambda_{\text{start}}^{(k)}, t_{\text{cut}}]$ and $[t_{\text{cut}}, \lambda_{\text{end}}^{(k)}]$.
17:         Add the two new sub-intervals to $\mathcal{I}_{\text{process}}$.
18:     **end if**
19: **end while**
20:                                           ▷ Refinement step: merge contours with too few eigenvalues.
21: Merge any contour in $\mathcal{C}_{\text{final}}$ containing fewer than $N_{min}$ eigenvalues with its nearest neighbor.
22: **return** Final contours $\mathcal{C}_{\text{final}}$.

---

**Algorithm 3** The Deepcontour Framework

---

1: **Input:** New system parameter function $a(\mathbf{x})$; GEP matrices $A, B$; trained ENO $\mathcal{G}_\theta$; KDE contour parameters $N_{\min}, N_{\max}, w$.
2: **Output:** Eigenpairs $\{(\lambda_j, \mathbf{x}_j)\}$.
3: *Stage 1: Rapid Spectral Prediction*
4: Discretize input function: $a_{\text{grid}} \leftarrow \text{Discretize}(a(\mathbf{x}))$.
5: Predict approximate spectrum using the trained ENO: $\hat{\Lambda} \leftarrow \mathcal{G}_\theta(a_{\text{grid}})$.
6: *Stage 2: Automated Contour Construction*
7: Generate a set of optimized contours by invoking the KDE-based process:
8: $\{\Gamma_k\}_{k=1}^{N_\gamma} \leftarrow$ Contour Construction$(\hat{\Lambda}, N_{\min}, N_{\max}, w)$ using Algorithm 2.
9: *Stage 3: Final CI Solve*
10: Pass the generated contours $\{\Gamma_k\}$ and matrices $A, B$ to the CI solver (e.g., CIRR or FEAST).
11: Compute the final eigenpairs $\{(\lambda_j, \mathbf{x}_j)\}$ by executing the CI solver in parallel.
12: **return** Computed eigenpairs.

---

## E. From Partial Differential Equation to GEPs

The large-scale GEPs of the form $A\mathbf{x} = \lambda B\mathbf{x}$ studied in this work are not abstract mathematical objects; they are the discrete representations of continuous physical systems. These systems are typically governed by Partial Differential Equations (PDEs) that describe phenomena such as vibration, heat diffusion, or quantum mechanics (Saad, 2011; Kressner, 2005). To solve these problems numerically, the continuous PDE must be transformed into a finite-dimensional matrix problem

through a process called discretization.

A common and powerful method for this is the Finite Element Method (FEM) (Bathe, 2006). In this approach, the physical domain is first partitioned into a fine mesh of smaller elements. The continuous solution field (e.g., displacement or temperature) is then approximated as a linear combination of basis functions (or shape functions) defined over these elements. Applying this approximation and the principles of variational calculus (specifically, deriving the weak form of the PDE) transforms the original differential operators into discrete, sparse matrices. Typically, the operator terms related to spatial derivatives (e.g., stiffness, conductivity) form the matrix $A$, while terms related to time derivatives or material capacity (e.g., mass, permittivity) form the matrix $B$. The dimension of these matrices, $N$, is determined by the number of degrees of freedom in the mesh.

The specific governing PDEs and the resulting GEP formulations for each of the five scientific domains analyzed in our experiments are provided in detail in Appendix F.3.

# F. Details of Experiment

## F.1. Specific parameters of the main experiment

**ENO Model Training and Prediction.** The ENO consists of an FNO-based backbone (Li et al., 2020) and an MLP-based prediction head (Bengio et al., 2017). The FNO backbone is composed of 4 Fourier layers with a uniform channel width of 64, and 20 modes are retained in each layer. The input function $a(\mathbf{x})$ is discretized onto a uniform grid before being processed by the network. The MLP head consists of 3 fully-connected layers with GELU activation and 128 hidden-dim. The model is trained end-to-end by minimizing the Mean Squared Error (MSE) loss between the predicted and ground-truth eigenvalues (Bengio et al., 2017). We used the Adam optimizer with a learning rate of $1 \times 10^{-3}$ and a batch size of 16. The training was conducted for 200 epochs on the GPU. We provide training time and training curves in Table 4 and Figure 5.

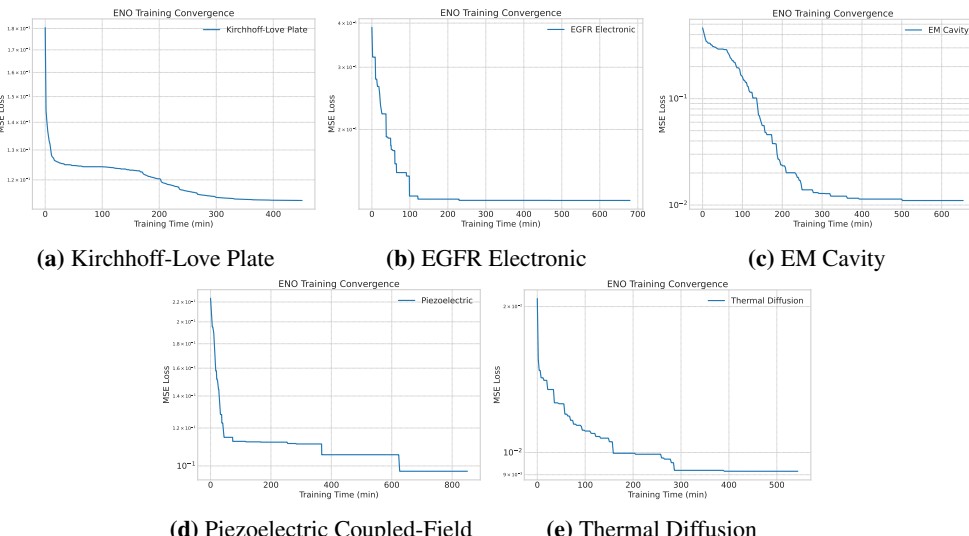

(a) Kirchhoff-Love Plate      (b) EGFR Electronic      (c) EM Cavity

(d) Piezoelectric Coupled-Field      (e) Thermal Diffusion

**Figure 5.** Training curves of our ENO model across five datasets.

**Table 4.** Total training time (in hours) for the ENO model over 200 epochs for each of the five scientific datasets.

| Dataset | Training Time (h) |
|---|---|
| Kirchhoff-Love Plate | 7.6 |
| EGFR Electronic | 10.89 |
| EM Cavity | 11.07 |
| Piezoelectric Coupled-Field | 14.21 |
| Thermal Diffusion | 9.21 |

**Table 5.** Hyperparameters of neural spectral predictors used in the ablation studies.

| Hyperparameter | ENO | DeepONet | MLP |
|---|---|---|---|
| Input encoding | Grid function | Grid samples + index embedding | Flattened grid |
| Layer | 4 | 4 / 3 | 4 |
| Width | 64 | 128 | 256 |
| Activation | GELU | GELU | GELU |
| Optimizer | Adam | Adam | Adam |
| Learning rate | $1 \times 10^{-3}$ | $1 \times 10^{-3}$ | $1 \times 10^{-3}$ |
| Batch size | 16 | 16 | 16 |
| Epochs | 200 | 200 | 200 |

**Neural baseline configurations.**    For fair comparison, all learning-based spectral predictors are trained with the same train/validation/test split, MSE loss, Adam optimizer, learning rate $1 \times 10^{-3}$, batch size 16, and 200 epochs. The architecture configurations are summarized in Table 5.

**KDE-based Contour Construction.**    Our automated contour construction follows the iterative process detailed in Algorithm 2. The core component is the interval sparsity function, $G_k(t)$, defined in Eq. (6). The hyperparameters for this process, which have been noted to have low sensitivity in similar spectral estimation tasks (Lin et al., 2016; Senzaki et al., 2010), were set with minimal tuning as follows: the maximum number of eigenvalues per contour $N_{max}$ was set to 50, the minimum number $N_{min}$ was set to 10, and the gap detection weight $w$ was set to 10. The threshold for identifying significant gaps was determined dynamically, as described in Algorithm 2.

**Safety-margin calibration.**    We use two safety buffers in contour construction: a local inflation factor $\alpha$ and a global spectral-hull buffer coefficient $\beta$. The local factor $\alpha$ controls the radius inflation of each contour, while $\beta$ determines the global buffer

$$\epsilon = \beta \left( \max(\hat{\Lambda}) - \min(\hat{\Lambda}) \right).$$

Both parameters are calibrated only on the validation set and then fixed for all test instances. Specifically, we select the smallest pair of $(\alpha, \beta)$ that keeps the validation contours covering the target eigenvalues under the prescribed residual tolerance. Among feasible pairs, we choose the one with the lowest average end-to-end time. No test-set eigenvalue information is used for selecting these parameters.

**Baseline Configuration.**    The choice of the number of scout iterations, $k$, represents a critical trade-off between the scout's efficiency and its predictive accuracy. A small number of iterations (e.g., $k < 30$) would reduce the scouting cost but yield a highly inaccurate spectral map, requiring an excessively large safety margin that degrades the final solver's performance. Conversely, a large number of iterations (e.g., $k > 100$) would produce a more accurate map, but the increase in the scouting time itself would offer diminishing returns, nullifying the cost-saving purpose of the two-stage approach (Saad, 2011). Therefore, we selected a fixed value of $k = 60$ as a balanced choice representing a strong configuration for the baseline methods. To ensure a fair comparison and reflect a realistic use-case, we did not perform extensive per-method tuning for this parameter. We also observed that performance for most methods was stable around this value. Following the scouting step, the safety margin for the rectangular contour was precisely calibrated for each baseline to be the minimum size required to enclose 100% of the target eigenvalues, a process critical for solver robustness (Gavin & Polizzi, 2016). See more details in Appendix F.2.

**CI Solver and Computing Infrastructure.**    The final high-fidelity solve was performed using the CIRR (Sakurai & Tadano, 2007) and FEAST (Polizzi, 2009) solvers, with results for CIRR presented in the main text and FEAST in Appendix F.1. All solves were conducted for five different accuracy tolerances, ranging from $10^{-2}$ to $10^{-12}$. Experiments were run on a compute node equipped with an Intel Xeon Gold 6248R CPU and an NVIDIA RTX4090 GPU. The deep learning components utilized the GPU, while the CI solvers and all baseline methods were executed in parallel on the CPU, leveraging all 20 available physical cores via OpenMP.

## F.2. Baseline Methods: Scouting-based Localization

In our experiments, we compare Deepcontour against the powerful strategy for contour construction: Scouting-based Localization (Saad, 2011; Gavin & Polizzi, 2016; Sakurai & Tadano, 2007). This is a two-stage "scout-then-solve" process designed to define a suitable integration contour for the final CI solver. This section details the specific procedures and configurations used for these baselines to ensure a fair and rigorous comparison.

### F.2.1. BASELINE CONTOUR CONSTRUCTION

The primary goal of the scouting stage is to obtain a rough estimate of the target eigenvalue locations. In our experimental setup, the objective is to find the $M$ smallest (in magnitude) eigenvalues of the GEP. Since standard Krylov subspace methods like Lanczos and Arnoldi are primarily designed for computing extremal eigenpairs (i.e., the largest eigenvalues) (Lehoucq et al., 1998), to force these methods to find the smallest eigenvalues, a standard and necessary approach is the shift-and-invert technique. Instead of applying the iterative solver to the original matrix pencil $(A, B)$, it is applied to the inverted operator $(A - \sigma B)^{-1}B$ with a shift $\sigma$ set to zero. The largest eigenvalues of this shift-and-invert operator correspond precisely to the smallest (in magnitude) eigenvalues of the original GEP. This process is computationally expensive as it requires a matrix factorization (e.g., LU decomposition) of the matrix $A$. The $k$ iterations are sufficient to produce a cluster of Ritz values that serves as a coarse estimate of the desired smallest eigenvalues. Based on the coarse distribution of Ritz values from the scouting stage, a integration contour is constructed for the CI solver. First, a tight-fitting axis-aligned bounding box (a rectangular contour) is computed that encloses all the generated Ritz values. However, this initial tight box is insufficient for two critical reasons. (1) **Enclosure of True Eigenvalues**: The Ritz values are only approximations. To guarantee that the contour encloses all the corresponding true eigenvalues, this initial box must be expanded by a safety margin factor (Gavin & Polizzi, 2016). For each baseline, this factor was precisely calibrated to the minimum size required to ensure 100% capture of the target eigenvalues. For instance, for the Arnoldi-Scout on the Kirchhoff-Love Plate dataset, this resulted in an expansion of the bounding box area by **69%** to guarantee full coverage. (2) **Numerical Stability and Efficiency**: The CI involves the resolvent $(zB - A)^{-1}$. If the contour path $z$ is positioned too close to a true eigenvalue on the real axis, the matrix $(zB - A)$ becomes nearly singular, which can lead to significant quadrature error and jeopardize numerical accuracy (Polizzi, 2009). Therefore, a 2D contour in the complex plane that maintains a safe distance is required. Additionally, to maintain quadrature efficiency, it is common practice to avoid contours with extreme aspect ratios (Bathe, 2006). In our experiments, we ensured the aspect ratio of the rectangular contours did not exceed 5, as a very elongated shape would require a prohibitively large number of quadrature points on its longest sides (Bathe, 2006).

### F.2.2. MORE PRACTICAL DETAILS

The final result of the above process is a single, relatively large, and conservative rectangular contour that is guaranteed to contain all target eigenvalues. **In fact, we also explored a more sophisticated clustering-based approach for the baseline. This strategy involves expanding each Ritz value into an interval with a width proportional to the largest spectral gap, merging any overlapping intervals, and then constructing a rectangular bounding box around each resulting cluster.** However, due to the significant uncertainty in the Ritz values from the scout, the required interval width was so large that all intervals invariably merged into a single cluster. This resulted in one large bounding box, offering no advantage over the simpler method. It is noteworthy that this same rule is used in our main ablation study (Table 2), where the highly accurate predictions from our ENO model allow it to successfully identify multiple, distinct contours. This highlights that the limitation lies not in the contouring rule itself, but in the low precision of the initial scouting-based prediction. **Similarly, we found that attempting to use a single large circular contour, despite its potential for higher quadrature efficiency, was also suboptimal; the coarse and scattered nature of the Ritz values resulted in a contour with an excessively large area compared to the tighter bounding box.** This highlights that the limitation lies not in the contouring rule itself, but in the low precision of the initial scouting-based prediction.

**Comparison with Deepcontour: From Estimation to Prediction**  The advantage of our Deepcontour framework lies in its fine-grained approach, benefiting from more accurate prior knowledge of spectral distribution. Instead of generating a single, large contour, based on high accuracy of ENO prediction, our KDE-based method identifies the natural spectral gaps in the predicted eigenvalue distribution. This allows it to partition the spectrum and construct multiple, smaller, and more tight-fitting circular contours. These smaller, localized contours result in significantly smaller projected problems for the CI solver, which is the primary reason for the superior *CI Solver Time Speedup* observed in our results.

### F.2.3. IMPLEMENTATION OF SCOUTING SOLVERS

In our implementation, we utilized five state-of-the-art iterative algorithms as scouts: the Arnoldi and Lanczos iterations (Saad, 2011), the more robust Krylov-Schur algorithm (Stewart, 2002), the Jacobi-Davidson method (Sleijpen & Van der Vorst, 2000), and a Riemannian optimization-based Gradient Descent method (Absil et al., 2008). Each solver is run for a fixed number of iterations ($k = 60$) on the shift-and-invert operator. The specific implementation of all solvers is handled by the PETSc (Balay et al., 2019) and SLEPc (Hernandez et al., 2005) (version 3.23.4) libraries.

### F.2.4. OVERVIEW OF ITERATIVE ALGORITHMS

Here we briefly introduce the core principles of the iterative algorithms used as scouts in our baselines.

**Arnoldi and Lanczos Iterations** These are foundational Krylov subspace methods (Saad, 2011). They construct an orthonormal basis for the Krylov subspace and solve the original GEP by projecting it onto this much smaller subspace. The Lanczos iteration is a highly efficient specialization for Hermitian problems that uses a short three-term recurrence, while the Arnoldi iteration is its more general counterpart for non-Hermitian matrices.

**Krylov-Schur Algorithm** This algorithm is an enhancement of the Arnoldi/Lanczos methods, designed for improved robustness and efficiency (Stewart, 2002). Its primary contribution is an elegant and numerically stable restarting mechanism, which allows the size of the Krylov subspace to be kept fixed, thus saving memory and computational cost without sacrificing the numerical quality.

**Jacobi-Davidson (JD) Method** The Jacobi-Davidson method is a subspace expansion technique that iteratively refines an approximate eigenpair (Sleijpen & Van der Vorst, 2000). At each step, it solves a "correction equation" to find the optimal update to the current solution. Its main strength lies in its ability to effectively incorporate preconditioning, making it very powerful for finding specific interior eigenvalues if a good preconditioner is available.

**Riemannian Gradient Descent (GD)** This approach reframes the eigenvalue problem as an optimization task (Absil et al., 2008). It seeks the eigenvectors by performing a gradient descent to minimize the Rayleigh quotient on a specific matrix manifold (the space of matrices with orthonormal columns). The iterative updates follow the curvature of this manifold, representing a distinct geometric approach to the problem.

### F.3. Datasets

For each of the five physical problems, we generated a dataset consisting of 1500 unique samples. Each sample corresponds to a different realization of the system's governing physical parameters (e.g., material density or thermal conductivity). In this work, we focus on Hermitian GEPs, which are foundational in many physical systems and are guaranteed to have real-valued eigenvalues. The datasets were split into 1000 samples for training and 500 samples for testing. For each sample, the ground-truth eigenvalues were pre-computed using the *Krylov-Schur algorithm*, as implemented in the SLEPc library (Hernandez et al., 2005), configured with a stringent convergence tolerance of $10^{-12}$.

The generation of ground-truth labels for the entire dataset of 1,500 samples is performed as a one-time offline pre-computation. On our hardware, this process takes approximately 7 minutes in total, averaging ∼0.28 seconds per sample using the high-precision Krylov-Schur algorithm configured with a tolerance of $10^{-12}$. While this pre-computation is efficient, it remains significantly slower than our ENO inference, which provides a spectral estimate for all samples in less than 0.01 seconds. The offline training time, ranging from 7.6 to 14.2 hours depending on the domain, represents a one-time investment that enables the rapid, data-informed contour construction of the Deepcontour framework in real-time applications.

### F.3.1. KIRCHHOFF-LOVE PLATE VIBRATION ANALYSIS

This dataset involves simulating the natural frequencies and modes of a thin plate structure, a classic problem in structural mechanics (Leissa, 1969; Reddy, 2006). The problem originates from the free vibration PDE for a Kirchhoff-Love plate:

$$D\Delta^2 w(\mathbf{x}, t) = \rho h \frac{\partial^2 w}{\partial t^2}(\mathbf{x}, t), \tag{12}$$

where $w$ is the transverse displacement, $D$ is the bending rigidity, $\rho$ is the material density, and $h$ is the plate thickness. By assuming a harmonic solution $w(\mathbf{x}, t) = \phi(\mathbf{x}) \cos(\omega t)$, we obtain the steady-state eigenvalue problem. To facilitate a finite-element discretization, this 4th-order PDE is reformulated using a mixed variable approach. By introducing an

intermediate variable $\psi = \Delta\phi$, the problem is decomposed into a system of two 2nd-order PDEs:

$$\begin{cases} \psi(\mathbf{x}) = \Delta\phi(\mathbf{x}) \\ D\Delta\psi(\mathbf{x}) = \rho h \omega^2 \phi(\mathbf{x}) \end{cases} \tag{13}$$

Discretizing this system using the Finite Element Method leads to the matrix GEP:

$$A\mathbf{u} = \lambda M\mathbf{u}, \tag{14}$$

where $A$ is the stiffness matrix assembled from the discretized Laplacian operators, $M$ is the mass matrix, $\mathbf{u}$ is the vector of nodal displacements, and the eigenvalue $\lambda = \omega^2$ corresponds to the square of the structure's natural frequency. In our dataset, the material density $\rho(\mathbf{x})$ was generated as a spatially varying function using Gaussian Random Fields (GRF) to simulate non-uniform materials, while other parameters were held constant.

### F.3.2. EGFR ELECTRONIC STRUCTURE CALCULATION

This dataset represents a typical class of GEPs in quantum chemistry, computing the electronic structure of the Epidermal Growth Factor Receptor (EGFR) molecule (Szabo & Ostlund, 2012; Roothaan, 1951). The problem is governed by the Hartree-Fock-Roothaan equations, a formulation of the time-independent Schrödinger equation for a single electron orbital $\psi_i(\mathbf{r})$:

$$\left(-\frac{1}{2}\nabla^2 + V_{\text{eff}}(\mathbf{r})\right)\psi_i(\mathbf{r}) = \epsilon_i \psi_i(\mathbf{r}), \tag{15}$$

where $\psi_i$ is the $i$-th molecular orbital (eigenfunction), $\epsilon_i$ is its corresponding energy (eigenvalue), and $V_{\text{eff}}$ is the effective potential experienced by the electron, which includes nuclear attraction and average electron-electron repulsion.

To solve this problem computationally, the continuous orbital functions $\psi_i$ are expanded in a discrete basis set. This standard procedure transforms the differential equation into the matrix GEP:

$$H\mathbf{c} = \lambda S\mathbf{c}, \tag{16}$$

where $H$ is the Hamiltonian matrix (often called the Fock matrix), which is the discretized form of the energy operator $(-\frac{1}{2}\nabla^2 + V_{\text{eff}})$; $S$ is the overlap matrix arising from the non-orthogonality of the basis functions; $\mathbf{c}$ is the vector of basis set coefficients for an eigenvector; and the eigenvalue $\lambda$ corresponds to the orbital energy $\epsilon_i$. Each sample in the dataset was generated by simulating different small perturbations to the molecular geometry, which in turn modifies the entries of the $H$ and $S$ matrices.

### F.3.3. ELECTROMAGNETIC CAVITY MODAL ANALYSIS

This problem involves solving for the eigenmodes of an electromagnetic resonator in the TE mode, which is widely used in RF engineering (Jin, 2015). For the Transverse Electric (TE) mode, the system simplifies to a scalar eigenvalue problem for the z-component of the electric field, $E_z(\mathbf{x})$:

$$\nabla \cdot \left(\frac{1}{\mu(\mathbf{x})}\nabla E_z(\mathbf{x})\right) + \omega^2 \epsilon(\mathbf{x}) E_z(\mathbf{x}) = 0, \tag{17}$$

where $\epsilon$ is the electric permittivity, $\mu$ is the magnetic permeability, and $\omega$ is the angular frequency. This PDE is solved using the Finite Element Method. By deriving the weak form and discretizing it with a suitable basis, we obtain the matrix GEP:

$$A\mathbf{u} = \lambda M\mathbf{u}, \tag{18}$$

where the entries of the stiffness matrix $A$ and mass matrix $M$ are given by the integrals over the basis functions $\phi_i, \phi_j$:

$$A_{ij} = \int_\Omega \frac{1}{\mu}\nabla\phi_j \cdot \nabla\phi_i \, d\mathbf{x}, \quad M_{ij} = \int_\Omega \epsilon\phi_i\phi_j \, d\mathbf{x}.$$

Here, $\mathbf{u}$ is the vector representing the discretized electric field, and the eigenvalue $\lambda = \omega^2$ is the square of the resonant angular frequency. In our dataset, we generated samples by creating spatially varying electric permittivity fields $\epsilon(\mathbf{x})$ using GRF to model an inhomogeneous medium, while the permeability $\mu$ was held constant.

### F.3.4. PIEZOELECTRIC COUPLED-FIELD MODAL ANALYSIS

This dataset analyzes the acoustic resonance modes in a 2D enclosed space, a fundamental problem in acoustics design that involves the coupling between mechanical and electrical fields (Tiersten, 1969; Allik & Hughes, 1970). The problem is governed by a set of coupled Partial Differential Equations (PDEs) representing mechanical motion and electrostatics:

$$\rho \frac{\partial^2 u_i}{\partial t^2} = \nabla_j \sigma_{ij}, \quad \nabla_i D_i = 0, \tag{19}$$

where $\mathbf{u}$ is the mechanical displacement, $\rho$ is the material density, $\sigma$ is the stress tensor, and $D$ is the electric displacement. The fields are linked via the piezoelectric constitutive relations. Assuming a harmonic solution ($\mathbf{u}(\mathbf{x}, t) = \mathbf{u}(\mathbf{x})e^{i\omega t}$ and $\phi(\mathbf{x}, t) = \phi(\mathbf{x})e^{i\omega t}$), and discretizing the system with the Finite Element Method results in the block matrix GEP:

$$\begin{bmatrix} K_{uu} & K_{u\phi} \\ K_{\phi u} & K_{\phi\phi} \end{bmatrix} \begin{bmatrix} \mathbf{u} \\ \boldsymbol{\phi} \end{bmatrix} = \omega^2 \begin{bmatrix} M_{uu} & 0 \\ 0 & 0 \end{bmatrix} \begin{bmatrix} \mathbf{u} \\ \boldsymbol{\phi} \end{bmatrix}, \tag{20}$$

where $K_{uu}$ is the structural stiffness matrix, $M_{uu}$ is the mass matrix, $K_{\phi\phi}$ is the dielectric stiffness matrix, and $K_{u\phi}$ and $K_{\phi u}$ are the piezoelectric coupling matrices. The solution consists of the eigenvector of discretized nodal displacements $\mathbf{u}$ and electric potentials $\phi$, and the eigenvalue $\lambda = \omega^2$, which is the square of the natural resonant frequency. We generated a diverse dataset by varying the geometric configuration and spatially-dependent material properties (e.g., elasticity tensor, piezoelectric tensor) for each sample.

### F.3.5. 2D THERMAL DIFFUSION MODAL ANALYSIS

This dataset investigates the stability and decay rates of thermal modes in a 2D system, which is critical to fields like fluid dynamics and thermal management (Gurtin, 1982; Ezekoye, 2016; Bathe, 2006). The problem originates from the time-dependent heat equation for a non-uniform medium:

$$c(\mathbf{x}) \frac{\partial T}{\partial t}(\mathbf{x}, t) = \nabla \cdot (k(\mathbf{x}) \nabla T(\mathbf{x}, t)), \tag{21}$$

where $T$ is the temperature, $k$ is the thermal conductivity, and $c$ is the heat capacity. To analyze the thermal modes, we assume a solution of the form $T(\mathbf{x}, t) = \hat{T}(\mathbf{x})e^{\lambda t}$, which transforms the PDE into a continuous eigenvalue problem. By deriving the weak form and discretizing it using the Finite Element Method, we obtain the matrix GEP:

$$K\mathbf{u} = \lambda C\mathbf{u}, \tag{22}$$

where the entries of the conductivity matrix $K$ (analogous to stiffness) and the capacity matrix $C$ (analogous to mass) are given by the integrals over the basis functions $\phi_i, \phi_j$:

$$K_{ij} = \int_\Omega k\nabla\phi_j \cdot \nabla\phi_i \, d\mathbf{x}, \quad C_{ij} = \int_\Omega c\phi_i\phi_j \, d\mathbf{x}.$$

Here, $\mathbf{u}$ is the vector representing the discretized temperature mode, and the eigenvalue $\lambda$ represents the decay rate of that mode. Each sample in our dataset was created by generating a different spatially varying thermal conductivity field $k(\mathbf{x})$ using the GRF method, while the heat capacity $c$ was held constant.

## G. Additional Experimental Results

### G.1. Performance Comparison with More Baselines

**Comparison with Feast** Table 6 presents the speedup comparison of our framework against the scouting-based baselines when using the FEAST algorithm as the underlying CI solver.

**Comparison with Iterative Methods** To provide a broader performance context, we extend the comparison from Figure 1 to include the standalone performance of the traditional iterative eigensolvers themselves. We use the five iterative algorithms (Arnoldi, Lanczos, etc.). Instead of running them for a fixed number of steps as a scout, they are configured as fully converged solvers, running until the final tolerance is met. To find the target smallest eigenvalues, these solvers are operated in a

**Table 6.** Speedup comparison of our Deepcontour framework against five scouting-based baselines using FEAST as CI solver. The results are shown for large-scale problems ($N = 50000$) across multiple datasets and for different accuracy tolerances. Each cell presents two metrics in the format: **End-to-End Speedup / CI Solver Time Speedup**.

| Dataset | Tolerance | vs. Arnoldi | vs. GD | vs. JD | vs. Lanczos | vs. KrylovSchur |
|---|---|---|---|---|---|---|
| Kirchhoff-Love Plate | 1e-2 | 4.99 / 3.79 | 3.75 / 3.64 | 3.74 / 3.57 | 2.51 / 2.09 | 2.50 / 1.78 |
| | 1e-4 | 4.24 / 4.10 | 3.84 / 3.61 | 3.63 / 3.12 | 2.89 / 1.97 | 2.35 / 1.70 |
| | 1e-7 | 4.38 / 4.07 | 3.80 / 2.64 | 3.79 / 2.63 | 2.23 / 1.60 | 2.09 / 1.58 |
| | 1e-10 | 4.23 / 3.18 | 4.09 / 3.15 | 3.77 / 2.79 | 2.07 / 1.70 | 2.06 / 1.71 |
| | 1e-12 | 3.00 / 3.28 | 2.99 / 2.32 | 2.98 / 2.31 | 2.24 / 1.63 | 2.10 / 1.59 |
| EGFR Electronic | 1e-2 | 4.59 / 2.79 | 3.79 / 2.78 | 3.78 / 2.69 | 1.98 / 1.93 | 1.97 / 1.92 |
| | 1e-4 | 3.92 / 2.84 | 2.92 / 2.17 | 2.72 / 2.16 | 2.14 / 1.83 | 2.13 / 1.51 |
| | 1e-7 | 3.00 / 2.57 | 2.99 / 2.31 | 2.57 / 1.71 | 2.07 / 1.67 | 1.95 / 1.66 |
| | 1e-10 | 2.96 / 2.54 | 2.76 / 1.82 | 2.75 / 1.83 | 1.85 / 1.53 | 1.78 / 1.37 |
| | 1e-12 | 2.53 / 1.73 | 2.24 / 1.72 | 2.23 / 1.71 | 1.68 / 1.70 | 1.67 / 1.58 |
| EM Cavity | 1e-2 | 3.29 / 3.16 | 3.28 / 2.71 | 2.97 / 2.09 | 2.26 / 1.69 | 2.21 / 1.68 |
| | 1e-4 | 2.86 / 2.26 | 2.85 / 2.25 | 2.59 / 2.02 | 2.20 / 1.96 | 2.13 / 1.80 |
| | 1e-7 | 2.86 / 2.24 | 2.79 / 2.23 | 2.25 / 2.09 | 2.09 / 1.66 | 1.68 / 1.65 |
| | 1e-10 | 3.19 / 2.07 | 2.18 / 1.86 | 2.17 / 1.85 | 1.97 / 1.74 | 1.82 / 1.73 |
| | 1e-12 | 2.44 / 2.16 | 2.43 / 2.15 | 2.42 / 1.97 | 1.96 / 1.70 | 1.82 / 1.56 |
| Piezoelectric Coupled-Field | 1e-2 | 3.18 / 2.82 | 2.47 / 2.63 | 2.46 / 2.08 | 1.99 / 1.82 | 1.98 / 1.81 |
| | 1e-4 | 2.79 / 2.19 | 2.43 / 2.18 | 2.42 / 2.07 | 2.32 / 1.65 | 2.09 / 1.64 |
| | 1e-7 | 2.78 / 2.14 | 2.77 / 1.99 | 2.31 / 1.82 | 2.30 / 1.54 | 1.98 / 1.53 |
| | 1e-10 | 2.87 / 2.04 | 2.86 / 1.80 | 2.61 / 1.79 | 2.53 / 1.78 | 1.65 / 1.53 |
| | 1e-12 | 2.82 / 1.80 | 2.81 / 1.79 | 2.12 / 1.76 | 1.99 / 1.75 | 1.52 / 1.56 |
| Thermal Diffusion | 1e-2 | 3.25 / 2.82 | 2.85 / 2.75 | 2.79 / 1.97 | 1.77 / 1.57 | 1.76 / 1.56 |
| | 1e-4 | 3.53 / 2.54 | 2.57 / 2.43 | 2.56 / 2.28 | 1.92 / 1.61 | 1.62 / 1.60 |
| | 1e-7 | 2.62 / 2.76 | 2.30 / 2.04 | 2.29 / 2.01 | 1.72 / 1.75 | 1.71 / 1.55 |
| | 1e-10 | 3.20 / 2.91 | 2.46 / 1.81 | 2.45 / 1.80 | 1.62 / 1.39 | 1.61 / 1.38 |
| | 1e-12 | 2.79 / 2.72 | 2.43 / 2.05 | 2.21 / 2.02 | 1.80 / 1.50 | 1.72 / 1.48 |

shift-and-invert mode. Figure 6 plots the convergence curve of Deepcontour, the scout-based CI methods, and these fully converged iterative solvers. The results reveal a three-tiered performance hierarchy. On our specific computational platform, the standalone iterative solvers are significantly slower than all CI-based approaches. The scouting-based methods occupy the middle ground, demonstrating the inherent efficiency of the CI paradigm. Our Deepcontour framework is the fastest, showcasing the substantial additional speedup gained by resolving the contour selection bottleneck. This result quantitatively validates the two primary motivations for our work.

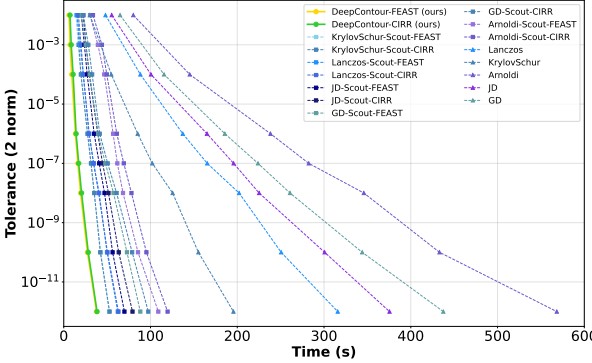

**Figure 6.** Performance comparison of Deepcontour, scouting-based CI methods, and standalone iterative solver. The test was performed on a curated set of representative large-scale problem instances ($N = 50000$), with instances drawn from each of our five scientific domains. The clear separation in performance validates the advantage of CI methods for this task and the further acceleration achieved by our framework.

## G.2. Analysis of Generated Contours

This section provides a more detailed analysis to demonstrate the superiority of our Deepcontour framework. We compare our method against Arnoldi-Scout baseline from two perspectives: the efficiency of the generated contours and the accuracy of the underlying spectral prediction.

### G.2.1. QUANTITATIVE COMPARISON OF CONTOUR EFFICIENCY

**Table 7.** Quantitative comparison of the total contour area for the Arnoldi-Scout and Lanczos-Scout baselines versus our Deepcontour framework on the $N = 50000$ scale. The ratio highlights the significant reduction in contour size achieved by our method.

| Dataset | Area Ratio (vs. Arnoldi) | Area Ratio (vs. Lanczos) |
|---|---|---|
| Kirchhoff-Love Plate | 6.5x | 4.8x |
| EGFR Electronic | 5.6x | 4.9x |
| EM Cavity | 4.1x | 2.9x |
| Piezoelectric | 4.6x | 3.5x |
| Thermal Diffusion | 3.8x | 3.1x |

A primary claim of our work is that Deepcontour generates more computationally efficient contours. A simple and direct measure of a contour's efficiency is its total area in the complex plane; a smaller area generally corresponds to a smaller projected problem and thus a faster CI solve time. Table 7 compares the total area of the final, safety-margin-adjusted contour generated by the Arnoldi-Scout baseline against the sum of the areas of the multiple, tight-fitting contours generated by our Deepcontour framework. Our framework consistently produces contours with a total area that is much smaller than those from the scouting-based baseline. This significant reduction in the integration domain is a direct result of our KDE-based approach, which can precisely partition the spectrum. This finding quantitatively explains the substantial *CI Solver Time Speedup* reported in the main experiments. Furthermore, as previously discussed, the circular contours generated by our method are inherently efficient for numerical quadrature, as the trapezoidal rule is known to exhibit exponential convergence on such domains (Polizzi, 2009; Sakurai & Tadano, 2007).

### G.2.2. COMPARISON OF SPECTRAL ESTIMATION ACCURACY

The superior quality of our contours stems from the high accuracy of our initial spectral prediction. To validate and intuitively demonstrate this, we quantitatively compare our ENO model's accuracy against the rough estimate from the Arnoldi-Scout. We use the Normalized Mean Squared Error (NMSE) as the metric, which is computed on standardized eigenvalue sets (zero mean, unit variance) to ensure a fair comparison across datasets with different physical scales. For our ENO, the NMSE is calculated directly on its $M$ predictions, while for the scout, it is calculated against the closest corresponding Ritz values from its generated subspace. As shown in Table 8, the spectral prediction from our ENO is consistently one to two orders of magnitude more accurate than the rough estimate provided by the scouting process. This high-accuracy prediction is the fundamental reason our KDE-based contouring is so effective at identifying real spectral gaps. Conversely, the low precision of the scout's estimate necessitates the use of a large, conservative safety margin, which inevitably leads to the oversized and inefficient contours quantified above.

## G.3. Extended Ablation Studies

### G.3.1. IMPLEMENTATION DETAILS OF NEURAL BASELINES AND ALTERNATIVE BACKBONES

We formulate spectral prediction as a function-to-vector regression task. Given a discretized physical parameter function $a_{\text{grid}}$, all learning-based baselines predict the same fixed-length target vector

$$\Lambda = (\lambda_1, \ldots, \lambda_M),$$

which contains the $M$ target eigenvalues. All models are trained using the same dataset split, MSE loss, and optimization protocol.

**MLP baseline.** For the MLP baseline, we use a non-operator regressor. The discretized input function $a_{\text{grid}}$ is flattened into a vector of size $d_a$, processed by four fully connected layers with GELU activation, and then mapped directly to an $M$-dimensional output. This baseline therefore learns a fixed-resolution mapping

$$a_{\text{grid}} \mapsto (\lambda_1, \ldots, \lambda_M),$$

**Table 8.** Normalized Mean Squared Error (NMSE) of the initial spectral prediction for the Arnoldi-Scout versus our ENO model on the $N = 50000$ scale.

| Dataset | Arnoldi-Scout MSE | ENO Prediction MSE |
|---|---|---|
| Kirchhoff-Love Plate | $9.8 \times 10^{-3}$ | $6.1 \times 10^{-5}$ |
| EGFR Electronic | $1.9 \times 10^{-3}$ | $5.5 \times 10^{-5}$ |
| EM Cavity | $8.8 \times 10^{-3}$ | $2.3 \times 10^{-4}$ |
| Piezoelectric | $5.2 \times 10^{-3}$ | $9.7 \times 10^{-5}$ |
| Thermal Diffusion | $7.4 \times 10^{-3}$ | $6.4 \times 10^{-4}$ |

without the spectral convolution or resolution-invariant parameterization used by FNO.

**DeepONet baseline.** For the DeepONet baseline (Lu et al., 2021), we adapt its dual-network structure to the spectral prediction task. The branch network encodes the sampled input function values on the same grid used by ENO. Since the output is an ordered eigenvalue vector rather than a spatial field, the trunk network takes the positional embedding of the target eigenvalue index $j \in \{1, \ldots, M\}$ as input. The branch and trunk features are combined and passed to a shared prediction head to output the corresponding eigenvalue $\hat{\lambda}_j$. The full predicted spectrum is obtained by evaluating the model over all target indices.

**Comparison with an alternative operator backbone.** To demonstrate the robustness of our hybrid framework and its compatibility with different neural operator backbones, we replace the FNO-based backbone (Li et al., 2020) in ENO with DeepONet while keeping the same training data, loss function, and optimization schedule. The evaluation is performed on the Kirchhoff–Love Plate dataset at the $N = 50000$ scale. We report the Normalized Mean Squared Error (NMSE) of spectral prediction, together with the resulting End-to-End and CI Solver Time Speedups against the KrylovSchur-Scout baseline under tolerance $10^{-7}$.

**Table 9.** Performance comparison of FNO and DeepONet backbones within the Deepcontour framework. Results are reported on the Kirchhoff–Love Plate dataset ($N = 50000$, tol=$10^{-7}$) against the KrylovSchur-Scout baseline. NMSE denotes the normalized spectral prediction error. E2E speedup and CI speedup denote end-to-end time speedup and CI solver time speedup, respectively.

| Backbone | NMSE | E2E Speedup | CI Speedup |
|---|---|---|---|
| FNO (Ours) | $8.12 \times 10^{-5}$ | $2.09\times$ | $1.59\times$ |
| DeepONet | $8.97 \times 10^{-5}$ | $1.97\times$ | $1.45\times$ |

As shown in Table 9, both operator backbones provide clear improvements over the scouting-based baseline, indicating that the proposed learning-guided contouring pipeline is not tied to a single neural operator architecture. The FNO backbone achieves slightly lower prediction error and higher speedups than DeepONet in this setting, supporting our choice of FNO as the default feature extractor for ENO.

**Table 10.** Ablation studies on the FNO backbone's hyperparameters.

| Hyperparameter | Value | MSE | E2E Speedup |
|---|---|---|---|
| Layer | 2 | $1.51 \times 10^{-1}$ | 1.71x |
| | 4 | $8.48 \times 10^{-2}$ | 1.95x |
| | 6 | $9.63 \times 10^{-2}$ | 1.91x |
| Width | 32 | $4.75 \times 10^{-2}$ | 1.84x |
| | 64 | $3.32 \times 10^{-2}$ | 1.95x |
| | 128 | $4.81 \times 10^{-2}$ | 1.76x |
| Mode | 12 | $6.13 \times 10^{-2}$ | 1.89x |
| | 16 | $6.13 \times 10^{-2}$ | 1.92x |
| | 20 | $3.79 \times 10^{-2}$ | 1.95x |

G.3.2. TRADITIONAL SCOUTS WITH KDE

To isolate and validate the critical contribution of our high-accuracy ENO predictor, we conducted an ablation study where we replaced the ENO module with a traditional scouting method, while retaining our KDE-based contour construction pipeline. This creates a strong hybrid baseline, termed "Scout+KDE," which allows us to test whether our automated

contouring logic alone is sufficient for top performance. We used the most robust scout, KrylovSchur, for this comparison. The evaluation was performed on 100 instances from the Kirchhoff-Love Plate dataset ($N = 50000$) with a CI solver tolerance of $10^{-7}$. We report the average number of missed eigenvalues (a measure of reliability) and the end-to-end solve time (a measure of overall efficiency). The results are provided in Table 11. The Scout+KDE baseline, despite leveraging our advanced KDE contouring module, performs significantly worse than the complete Deepcontour framework. It fails to reliably capture all target eigenvalues (missing nearly 42 on average). This performance degradation occurs because the low-precision Ritz values generated by the scout provide a noisy and unreliable input to the KDE module, leading to suboptimal partitioning and inefficient contours. This experiment confirms that the remarkable efficiency of Deepcontour is not just due to the automated KDE pipeline, but is critically dependent on the high-accuracy spectral prediction provided by the ENO module. Notably, this reliability failure is not due to suboptimal KDE tuning; even when using more conservative weight parameters ($w \in [1, 10]$), the low-quality spectral prediction consistently led to missed eigenvalues, an effect further detailed in Section G.3.4.

**Table 11.** Ablation study comparing our full Deepcontour framework against a hybrid baseline that combines the KrylovSchur-Scout with our KDE pipeline (denoted as "KS-Scout+KDE"). The results highlight the importance of the ENO's high-accuracy prediction.

| Model | # of Missed Eigenvalues | Solve Time (s) |
|---|---|---|
| Deepcontour | 0 | 25.3 |
| KS-Scout+KDE | 41.8 | 41.7 |

### G.3.3. IMPACT OF FNO HYPERPARAMETERS

To validate our chosen FNO architecture, we conducted an ablation study on its three key hyperparameters: model layers, mode and width for fourier layer. We conduct experiments to investigate the impacts of these hyperparameters. The performance was evaluated on the Kirchhoff-Love Plate dataset ($N = 25000$) and measured by the predictive accuracy (MSE) and the resulting End-to-End Speedup against the KrylovSchur-Scout baseline (tolerance=$10^{-7}$).

### G.3.4. IMPACT OF KDE HYPERPARAMETERS

The key hyperparameter in our KDE-based contour construction is the weight $w$ in the interval sparsity function (Eq. (6)), which controls the sensitivity of the spectral gap detection. To demonstrate the robustness of our framework to this choice, we conducted an ablation study on the value of $w$. The experiment was performed on 100 instances from the Kirchhoff-Love Plate dataset ($N = 50000$) with a CI solver tolerance of $10^{-7}$.

The results in Table 15 show that our method is robust across a reasonable range of $w$ values. A very small weight ($w = 1$) makes the gap detection less sensitive, resulting in fewer, oversized contours and thus a less efficient CI solve. Conversely, a very large weight ($w = 50$) makes the process overly sensitive to small fluctuations in the predicted density, leading to incorrect partitioning and missed eigenvalues. Our chosen value of $w = 10$ provides an excellent balance, achieving perfect reliability with the highest efficiency. The stable performance in the range of $w \in [5, 20]$ confirms that our KDE module does not require extensive, problem-specific hyperparameter tuning.

### G.3.5. SENSITIVITY TO $N_{\min}$ AND $N_{\max}$

We further evaluate the sensitivity of Deepcontour to the eigenvalue-count parameters $N_{\min}$ and $N_{\max}$, which control the desired number of predicted eigenvalues enclosed by each contour. The experiment is conducted under the same setting as the main ablation study. As shown in Table 16, Deepcontour remains stable across a moderate range of values, and the default setting $(N_{\min}, N_{\max}) = (10, 50)$ achieves the best overall speedup.

A smaller $N_{\max}$ tends to generate more contours and increases contour-management overhead, whereas a larger $N_{\max}$ may produce larger projected subproblems. The default setting provides a good trade-off between contour compactness and solver overhead.

### G.3.6. COMPARISON WITH ALTERNATIVE SPECTRUM PARTITIONING STRATEGIES

To further examine the role of KDE in contour construction, we compare it with two lightweight spectrum-partitioning alternatives under the same downstream contour-generation protocol. The first baseline uses a direct gap-threshold split, which partitions the predicted spectrum whenever the spacing between two adjacent predicted eigenvalues exceeds a fixed

**Table 12.** Comparison of the CIRR solving time (in seconds) using contours generated by our Deepcontour framework versus five scouting-based baselines. The results, shown for large-scale problems ($N = 50000$), are presented as Mean $\pm$ Standard Deviation. This directly highlights the effectiveness of our contour generation strategy.

| Dataset | Tolerance | Deepcontour (Ours) | vs. Arnoldi | vs. GD | vs. JD | vs. Lanczos | vs. KrylovSchur |
|---|---|---|---|---|---|---|---|
| Kirchhoff-Love Plate | 1e-2 | 2.13 ± 0.11 | 10.01 ± 0.62 | 8.20 ± 0.45 | 7.97 ± 0.51 | 4.58 ± 0.23 | 4.45 ± 0.27 |
| | 1e-4 | 4.64 ± 0.25 | 19.82 ± 0.98 | 17.08 ± 0.81 | 15.00 ± 0.79 | 9.47 ± 0.41 | 9.33 ± 0.49 |
| | 1e-7 | 8.22 ± 0.41 | 32.71 ± 1.54 | 24.91 ± 1.12 | 25.89 ± 1.34 | 16.28 ± 0.78 | 16.28 ± 0.83 |
| | 1e-10 | 15.30 ± 0.75 | 58.75 ± 2.81 | 44.22 ± 2.15 | 43.91 ± 2.21 | 29.22 ± 1.40 | 29.68 ± 1.51 |
| | 1e-12 | 28.65 ± 1.38 | 99.70 ± 4.58 | 80.51 ± 3.97 | 84.52 ± 4.21 | 53.58 ± 2.65 | 53.58 ± 2.74 |
| EGFR Electronic | 1e-2 | 5.88 ± 0.29 | 19.01 ± 0.91 | 18.35 ± 0.88 | 16.76 ± 0.81 | 12.35 ± 0.60 | 11.88 ± 0.58 |
| | 1e-4 | 10.51 ± 0.51 | 32.06 ± 1.55 | 26.80 ± 1.30 | 28.80 ± 1.40 | 21.13 ± 1.02 | 19.44 ± 0.94 |
| | 1e-7 | 17.48 ± 0.85 | 43.18 ± 2.10 | 41.08 ± 2.00 | 35.13 ± 1.71 | 34.09 ± 1.66 | 31.29 ± 1.52 |
| | 1e-10 | 28.13 ± 1.37 | 63.57 ± 3.10 | 62.17 ± 3.03 | 57.67 ± 2.81 | 52.88 ± 2.58 | 47.82 ± 2.33 |
| | 1e-12 | 38.20 ± 1.86 | 79.84 ± 3.89 | 84.42 ± 4.11 | 73.04 ± 3.56 | 70.67 ± 3.44 | 69.91 ± 3.41 |
| EM Cavity | 1e-2 | 7.56 ± 0.38 | 20.94 ± 1.05 | 19.28 ± 0.96 | 18.75 ± 0.94 | 15.72 ± 0.79 | 15.35 ± 0.77 |
| | 1e-4 | 12.01 ± 0.60 | 31.71 ± 1.59 | 29.78 ± 1.49 | 29.00 ± 1.45 | 23.90 ± 1.20 | 23.30 ± 1.17 |
| | 1e-7 | 20.27 ± 1.01 | 47.84 ± 2.39 | 46.82 ± 2.34 | 44.59 ± 2.23 | 37.70 ± 1.88 | 38.31 ± 1.92 |
| | 1e-10 | 31.63 ± 1.58 | 71.17 ± 3.56 | 70.54 ± 3.53 | 66.74 ± 3.34 | 61.05 ± 3.05 | 58.49 ± 2.92 |
| | 1e-12 | 45.31 ± 2.27 | 106.93 ± 5.35 | 103.31 ± 5.17 | 96.96 ± 4.85 | 83.37 ± 4.17 | 77.93 ± 3.90 |
| Piezoelectric Coupled-Field | 1e-2 | 12.15 ± 0.61 | 34.51 ± 1.73 | 31.34 ± 1.57 | 29.77 ± 1.49 | 22.96 ± 1.15 | 23.94 ± 1.20 |
| | 1e-4 | 20.43 ± 1.02 | 51.08 ± 2.55 | 51.28 ± 2.56 | 48.62 ± 2.43 | 33.30 ± 1.67 | 38.41 ± 1.92 |
| | 1e-7 | 38.52 ± 1.93 | 80.14 ± 4.01 | 88.60 ± 4.43 | 86.67 ± 4.33 | 73.57 ± 3.68 | 70.50 ± 3.52 |
| | 1e-10 | 66.10 ± 3.31 | 146.74 ± 7.34 | 146.08 ± 7.30 | 141.45 ± 7.07 | 122.28 ± 6.11 | 115.68 ± 5.78 |
| | 1e-12 | 89.21 ± 4.46 | 195.37 ± 9.77 | 201.62 ± 10.08 | 194.48 ± 9.72 | 162.36 ± 8.12 | 161.47 ± 8.07 |
| Thermal Diffusion | 1e-2 | 4.08 ± 0.20 | 12.12 ± 0.61 | 10.12 ± 0.51 | 9.83 ± 0.49 | 7.63 ± 0.38 | 7.96 ± 0.40 |
| | 1e-4 | 8.59 ± 0.43 | 21.56 ± 1.08 | 20.53 ± 1.03 | 20.19 ± 1.01 | 15.46 ± 0.77 | 15.98 ± 0.80 |
| | 1e-7 | 15.40 ± 0.77 | 37.11 ± 1.86 | 35.73 ± 1.79 | 33.88 ± 1.69 | 27.57 ± 1.38 | 27.87 ± 1.39 |
| | 1e-10 | 25.71 ± 1.29 | 60.16 ± 3.01 | 56.05 ± 2.80 | 54.25 ± 2.71 | 43.19 ± 2.16 | 45.00 ± 2.25 |
| | 1e-12 | 37.32 ± 1.87 | 90.31 ± 4.52 | 82.48 ± 4.12 | 80.22 ± 4.01 | 66.44 ± 3.32 | 64.92 ± 3.25 |

**Table 13.** Comparison of the FEAST solving time (in seconds) using contours generated by our Deepcontour framework versus five scouting-based baselines. The results, shown for large-scale problems ($N = 50000$), are presented as Mean $\pm$ Standard Deviation. This directly highlights the effectiveness of our contour generation strategy.

| Dataset | Tolerance | Deepcontour (Ours) | vs. Arnoldi | vs. GD | vs. JD | vs. Lanczos | vs. KrylovSchur |
|---|---|---|---|---|---|---|---|
| Kirchhoff-Love Plate | 1e-2 | 1.92 ± 0.10 | 7.98 ± 0.35 | 7.00 ± 0.41 | 6.85 ± 0.39 | 4.01 ± 0.22 | 3.42 ± 0.18 |
| | 1e-4 | 4.33 ± 0.21 | 17.75 ± 0.88 | 15.63 ± 0.76 | 13.51 ± 0.69 | 8.53 ± 0.43 | 7.36 ± 0.37 |
| | 1e-7 | 7.58 ± 0.38 | 30.86 ± 1.51 | 20.01 ± 1.00 | 19.94 ± 1.02 | 12.13 ± 0.61 | 11.98 ± 0.60 |
| | 1e-10 | 13.25 ± 0.65 | 42.14 ± 2.10 | 41.74 ± 2.09 | 36.97 ± 1.85 | 22.52 ± 1.13 | 22.66 ± 1.15 |
| | 1e-12 | 24.87 ± 1.23 | 81.57 ± 4.01 | 57.70 ± 2.89 | 57.45 ± 2.87 | 40.54 ± 2.03 | 39.54 ± 1.98 |
| EGFR Electronic | 1e-2 | 5.13 ± 0.26 | 14.31 ± 0.72 | 14.26 ± 0.71 | 13.80 ± 0.69 | 9.90 ± 0.50 | 9.85 ± 0.49 |
| | 1e-4 | 9.29 ± 0.46 | 26.39 ± 1.32 | 20.16 ± 1.01 | 20.07 ± 1.00 | 16.08 ± 0.80 | 14.03 ± 0.70 |
| | 1e-7 | 17.05 ± 0.85 | 43.82 ± 2.19 | 39.39 ± 1.97 | 29.16 ± 1.46 | 28.47 ± 1.42 | 28.30 ± 1.41 |
| | 1e-10 | 27.60 ± 1.38 | 70.10 ± 3.50 | 50.23 ± 2.51 | 50.51 ± 2.53 | 42.23 ± 2.11 | 37.81 ± 1.89 |
| | 1e-12 | 36.98 ± 1.85 | 63.98 ± 3.20 | 63.61 ± 3.18 | 63.24 ± 3.16 | 62.87 ± 3.14 | 58.43 ± 2.92 |
| EM Cavity | 1e-2 | 5.93 ± 0.30 | 17.14 ± 0.94 | 16.07 ± 0.80 | 12.39 ± 0.62 | 10.02 ± 0.50 | 9.96 ± 0.50 |
| | 1e-4 | 9.74 ± 0.49 | 22.01 ± 1.10 | 21.92 ± 1.10 | 19.67 ± 0.98 | 19.09 ± 0.95 | 17.53 ± 0.88 |
| | 1e-7 | 18.38 ± 0.92 | 41.17 ± 2.06 | 40.99 ± 2.05 | 38.41 ± 1.92 | 30.51 ± 1.53 | 30.33 ± 1.52 |
| | 1e-10 | 30.68 ± 1.53 | 63.51 ± 3.18 | 57.07 ± 2.85 | 56.76 ± 2.84 | 53.38 ± 2.67 | 53.09 ± 2.65 |
| | 1e-12 | 43.04 ± 2.15 | 92.97 ± 4.65 | 92.54 ± 4.63 | 84.79 ± 4.24 | 73.17 ± 3.66 | 67.14 ± 3.36 |
| Piezoelectric Coupled-Field | 1e-2 | 11.44 ± 0.57 | 32.26 ± 1.61 | 30.09 ± 1.50 | 23.80 ± 1.19 | 20.82 ± 1.04 | 20.71 ± 1.04 |
| | 1e-4 | 19.26 ± 0.96 | 42.18 ± 2.11 | 42.00 ± 2.10 | 40.09 ± 2.00 | 31.78 ± 1.59 | 31.59 ± 1.58 |
| | 1e-7 | 37.08 ± 1.85 | 79.35 ± 3.97 | 73.79 ± 3.69 | 67.49 ± 3.37 | 57.10 ± 2.85 | 56.73 ± 2.84 |
| | 1e-10 | 64.83 ± 3.24 | 132.25 ± 6.61 | 116.69 ± 5.83 | 110.55 ± 5.53 | 115.39 ± 5.77 | 99.20 ± 4.96 |
| | 1e-12 | 88.95 ± 4.45 | 160.11 ± 8.01 | 159.22 ± 7.96 | 156.55 ± 7.83 | 155.66 ± 7.78 | 138.76 ± 6.94 |
| Thermal Diffusion | 1e-2 | 3.52 ± 0.18 | 9.93 ± 0.50 | 9.68 ± 0.48 | 6.93 ± 0.35 | 5.53 ± 0.28 | 5.49 ± 0.27 |
| | 1e-4 | 8.00 ± 0.40 | 20.32 ± 1.02 | 19.44 ± 0.97 | 18.24 ± 0.91 | 12.88 ± 0.64 | 12.80 ± 0.64 |
| | 1e-7 | 14.25 ± 0.71 | 39.33 ± 1.97 | 29.07 ± 1.45 | 28.64 ± 1.43 | 24.94 ± 1.25 | 22.10 ± 1.10 |
| | 1e-10 | 24.37 ± 1.22 | 70.92 ± 3.55 | 44.11 ± 2.21 | 43.87 ± 2.19 | 33.87 ± 1.69 | 33.64 ± 1.68 |
| | 1e-12 | 35.96 ± 1.80 | 97.81 ± 4.89 | 73.72 ± 3.69 | 72.64 ± 3.63 | 53.94 ± 2.70 | 53.22 ± 2.66 |

threshold. The second baseline uses $K$-means clustering on the one-dimensional predicted eigenvalues and then constructs contours around the resulting clusters. All methods use the same ENO predictions and the same CI eigensolver.

As shown in Table 17, KDE provides the best overall performance among the tested lightweight partitioning strategies. The gap-threshold split can maintain eigenvalue coverage, but its fixed threshold is less adaptive to non-uniform spectral density, leading to larger or less balanced contours. The $K$-means split is also less suitable for this task because it requires specifying the number of clusters and may place boundaries in regions that are not sufficiently sparse. In contrast, KDE

**Table 14.** Component-wise average runtime (in seconds) for the pre-computation stage. The total pre-computation for Deepcontour is the sum of ENO Inference and KDE Construction.

| Dataset | Deepcontour | | Scouting Baselines | |
|---|---|---|---|---|
| | ENO Inference | KDE Construction | Arnoldi-Scout | Lanczos-Scout |
| Kirchhoff-Love Plate | 0.0081s | 1.5s | 11.78s | 8.19s |
| EGFR Electronic | 0.0083s | 1.7s | 9.15s | 7.92s |
| EM Cavity | 0.0079s | 1.4s | 8.98s | 7.13s |
| Piezoelectric Coupled-Field | 0.0085s | 1.8s | 12.10s | 9.55s |
| Thermal Diffusion | 0.0078s | 1.3s | 8.55s | 6.95s |

**Table 15.** Ablation study on the KDE weight parameter $w$. The configuration used in our main experiments ($w = 10$) is highlighted in bold.

| Weight ($w$) | # of Missed Eigenvalues | CI Solver Time (s) |
|---|---|---|
| 1 | 0 | 35.1 |
| 5 | 0 | 27.8 |
| 10 | 0 | 25.3 |
| 20 | 0.5 | 24.1 |
| 50 | 11.2 | 28.4 |

identifies split points from local minima of the smoothed spectral density, which better matches the goal of placing contour boundaries in low-density spectral gaps.

### G.4. Super-resolution protocol and metrics

**Protocol (train-low, test-high; no finetuning).** We evaluate resolution invariance of the FNO-based ENO via a train-low/test-high protocol. ENO is trained only at $N_{\text{train}} = 12{,}500$ and directly deployed to $N_{\text{test}} \in \{25{,}000, 50{,}000\}$ without any finetuning. For each $N_{\text{test}}$, we keep the main setting unchanged by targeting $M = 0.01 N_{\text{test}}$ smallest-magnitude eigenvalues and sweeping solver tolerances from $10^{-2}$ to $10^{-12}$. All contour construction and CI-solver hyperparameters are identical to the main experiments.

**Metrics.** We report (i) $\text{NMSE}_{\text{spec}}$, the spectral estimation NMSE computed on standardized eigenvalue sets (zero mean, unit variance) using ENO's $M$ predictions, and (ii) end-to-end metrics including Miss (missing target eigenvalues), total wall-clock time $T_{\text{ours}}$, and speedup $S = T_{\text{base}}/T_{\text{ours}}$. We define $\text{Miss}_i = \max\{0, M - |\Lambda_i|\}$, where $\Lambda_i$ is the set of *unique* eigenvalues returned by the end-to-end pipeline for instance $i$.

**Baseline.** All results in this section use **CIRR** as the contour-integral eigensolver. To keep the super-resolution study focused and concise, we use a single fixed scouting baseline, **Scout+KDE** with **KrylovSchur scouting** and a fixed budget $k = 60$, consistent with our main protocol.

### G.5. Detailed Runtimes and Additional Metrics

We compare average time costs of our data-driven contour design (ENO inference and KDE construction) against the scouting stage of two representative baselines (Arnoldi-Scout and Lanczos-Scout) in Table 14. Table 14 presents the component-wise average runtime for five datasets ($N = 50000$) problems. The times were averaged over 100 distinct instances from each dataset. We present detailed runtime of CIRR and FEAST solving for each contour methods in Table 12 and Table 13.

### G.6. Robustness under Mild Distribution Shifts

Since Deepcontour uses ENO predictions as spectral priors, its performance may degrade when test instances deviate from the training distribution. However, the final eigenpairs are still computed and verified by the downstream CI eigensolver. Thus, moderate prediction errors mainly affect contour quality rather than bypassing the numerical correctness mechanism.

To evaluate this effect, we conduct a controlled mild extrapolation experiment. The model is trained on the original parameter range and tested on in-distribution instances, $+5\%$ extrapolated instances, and $+10\%$ extrapolated instances. We report

**Table 16.** Sensitivity analysis of the contour eigenvalue-count parameters $N_{\min}$ and $N_{\max}$. The default setting used in the main experiments is highlighted.

| $(N_{\min}, N_{\max})$ | E2E Speedup | CI Speedup |
|:---:|:---:|:---:|
| $(5, 25)$ | $2.39\times$ | $1.94\times$ |
| $(\mathbf{10}, \mathbf{50})$ | $\mathbf{2.48}\times$ | $\mathbf{1.98}\times$ |
| $(15, 75)$ | $2.34\times$ | $1.89\times$ |

**Table 17.** Comparison with alternative spectrum partitioning strategies. KDE achieves the best end-to-end efficiency while maintaining complete eigenvalue coverage.

| Method | Miss | CI Time (s) | E2E Time (s) |
|:---|:---:|:---:|:---:|
| KDE | 0.0 | 25.3 | 26.8 |
| Gap-threshold split | 0.0 | 30.9 | 32.4 |
| $K$-means split | 0.9 | 32.7 | 34.2 |

the spectral prediction error (NMSE_spec), the average number of missed target eigenvalues (Miss), and the end-to-end speedup over the scouting-based baseline.

As shown in Table 19, the spectral prediction error increases under extrapolation, but the degradation is gradual. Even under $+10\%$ extrapolation, Deepcontour keeps the average number of missed eigenvalues close to zero and maintains non-trivial acceleration. This suggests that the framework is robust to moderate prediction errors: less accurate spectral priors tend to produce slightly less efficient contours, rather than immediate solver failure.

In practice, robustness can be further improved by conservative safeguards. When the ENO prediction is unreliable or the predicted partition is unstable, the safety margin can be enlarged. After the CI solve, residual-based diagnostics, eigenvalue-count checks, and boundary-proximity checks can be used to detect potential contour failures. If needed, the contour can be expanded, re-partitioned, or replaced by a conservative scout-based contour.

### G.7. Quantitative Case Study for a Representative Instance

To provide a clear context for the observed speedups, we present a detailed comparison of Deepcontour and the Arnoldi-Scout baseline for a single large-scale instance from the Kirchhoff-Love Plate dataset ($N = 50,000, \text{tol} = 10^{-7}$). Table 15 consolidates metrics from the pre-computation and solving stages to illustrate the efficiency of our approach. All CI solve times in Table 20 are measured with **CIRR**.

This instance-level breakdown confirms that the high-accuracy spectral prediction of the ENO module allows the KDE pipeline to partition the spectrum into smaller, more efficient integration domains. This reduction in the total contour area directly leads to smaller projected subproblems and a substantial decrease in the high-fidelity solve time.

### G.8. Effect of Spectral Prediction Quality

We further study how the quality of ENO spectral prediction affects downstream contour construction and CI solving. This analysis complements the amortized training-cost discussion: here we focus on the inference-stage behavior under different levels of spectral prediction error. We use the validation error of ENO as the predictor-quality measure and report two downstream metrics: the end-to-end speedup over Arnoldi-Scout and the missed-eigenvalue rate. The dashed horizontal line indicates speedup $= 1$, below which the learning-guided contour no longer provides a positive efficiency gain.

As shown in Figure 7, the downstream performance improves as the ENO validation error decreases. When the validation error is small, Deepcontour achieves stable positive speedups with nearly zero missed eigenvalues. As the prediction error increases, the generated contours become less reliable and less compact, leading to reduced speedup and a higher miss rate. This trend is consistent on both the Kirchhoff–Love Plate and Piezoelectric Coupled-Field benchmarks.

These results clarify the role of ENO in Deepcontour. The predictor does not need to produce highly accurate eigenvalues to be useful; it mainly needs to capture the coarse spectral range and density distribution so that KDE can construct informative contours. However, if the prediction error becomes too large, the learning-guided contour may lose its advantage and can even fall below the scouting baseline. In such cases, the framework can enlarge the safety margin or fall back to a

**Table 18.** Super-resolution results (train-low, test-high; no finetuning). ENO is trained at $N_{\text{train}} = 12{,}500$ and evaluated at higher resolutions $N_{\text{test}} \in \{25{,}000, 50{,}000\}$. We set $M = 0.01 N_{\text{test}}$ and use tolerances from $10^{-2}$ to $10^{-12}$. Speedup is defined as $S = T_{\text{base}}/T_{\text{ours}}$, where $T_{\text{base}}$ is **Scout+KDE with Krylov–Schur scouting** using $k = 60$ iterations.

| Dataset | $N_{\text{train}} \to N_{\text{test}}$ | $\text{NMSE}_{\text{spec}}$ | Miss | $T_{\text{ours}}(s)$ | $S$ |
|---|---|---|---|---|---|
| Kirchhoff–Love Plate | $12500 \to 25000$ | 0.034 | 0.00 | 26.1 | 2.27 |
| Kirchhoff–Love Plate | $12500 \to 50000$ | 0.075 | 0.00 | 57.3 | 2.37 |
| EGFR Electronic | $12500 \to 25000$ | 0.037 | 0.00 | 23.9 | 1.98 |
| EGFR Electronic | $12500 \to 50000$ | 0.079 | 0.00 | 50.2 | 2.01 |
| EM Cavity | $12500 \to 25000$ | 0.036 | 0.00 | 21.0 | 1.92 |
| EM Cavity | $12500 \to 50000$ | 0.081 | 0.00 | 45.7 | 1.95 |
| Piezoelectric | $12500 \to 25000$ | 0.046 | 0.05 | 32.0 | 1.76 |
| Piezoelectric | $12500 \to 50000$ | 0.088 | 0.09 | 70.6 | 1.85 |
| Thermal Diffusion | $12500 \to 25000$ | 0.044 | 0.03 | 20.4 | 1.71 |
| Thermal Diffusion | $12500 \to 50000$ | 0.093 | 0.12 | 42.6 | 1.79 |

**Table 19.** Robustness of Deepcontour under mild distribution shifts. The model is trained on the original parameter range and tested on in-distribution and extrapolated ranges. Deepcontour shows gradual degradation as the shift increases, while maintaining nearly complete eigenvalue coverage and non-trivial end-to-end acceleration.

| Domain | Shift | NMSE_spec | Miss | Speedup |
|---|---|---|---|---|
| Plate | ID | 0.031 | 0.00 | 3.70 |
| Plate | $+5\%$ extrap. | 0.038 | 0.00 | 3.56 |
| Plate | $+10\%$ extrap. | 0.046 | 0.01 | 3.37 |
| Piezo | ID | 0.042 | 0.00 | 3.05 |
| Piezo | $+5\%$ extrap. | 0.050 | 0.01 | 2.96 |
| Piezo | $+10\%$ extrap. | 0.061 | 0.03 | 2.84 |

conservative scout-based contour. This confirms that Deepcontour benefits from accurate spectral priors, while its practical deployment should monitor prediction reliability.

## H. Limitations and Future Work

Despite these promising results, our framework has limitations that open avenues for future research. The predictive accuracy of the ENO is contingent on the diversity of the training data, and its generalization to physical systems far outside the training distribution remains to be explored. Furthermore, while our method is applicable in principle to non-Hermitian problems, this work focused on the real-valued spectra of Hermitian systems. Future work could therefore extend the framework to handle complex spectra, possibly by employing 2D KDE. Investigating active learning strategies to reduce data dependency and applying this hybrid "predict-then-guide" philosophy to other challenging numerical tasks, such as nonlinear eigenvalue problems, are also exciting directions for further research.

**Table 20.** Comparative metrics for a Kirchhoff-Love Plate instance ($N = 50,000$ at $10^{-7}$ tolerance).

| Metric | Deepcontour (Ours) | Arnoldi-Scout (Baseline) |
|---|---|---|
| Contour Geometry | Multiple Circles | Single Rectangular Box |
| Total Contour Area Ratio | **1.0× (Ref.)** | 6.5× |
| Pre-computation Time (s) | **1.51** (ENO+KDE) | 11.78 (Scouting) |
| Avg. Projection Dimension ($r$) | 52 | 614 |
| Quadrature Nodes per Contour | 32 | 128 |
| CI Solve Time (s) | **17.24** | 57.71 |
| Total End-to-End Time (s) | **18.75** | 69.49 |

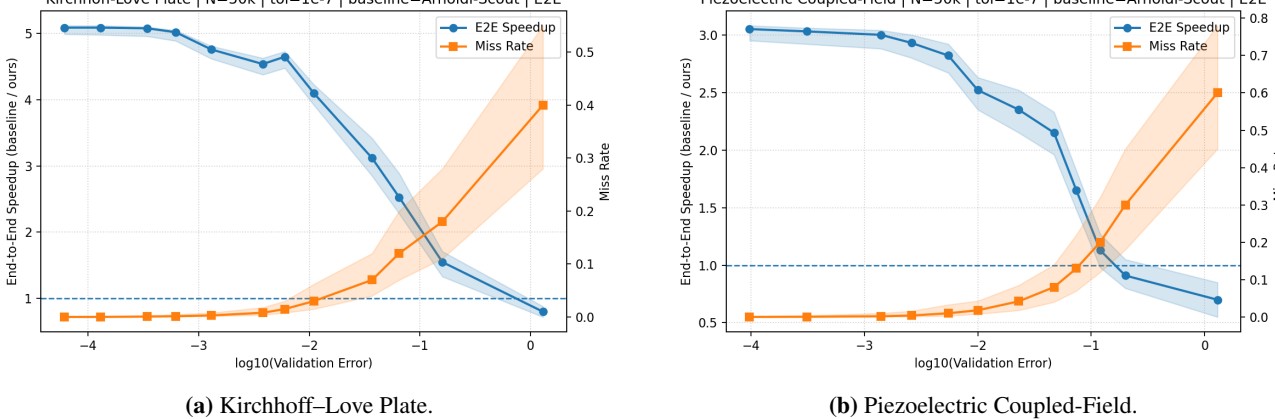

**(a)** Kirchhoff–Love Plate.

**(b)** Piezoelectric Coupled-Field.

**Figure 7.** Effect of ENO prediction quality on downstream performance. Both experiments use $N = 50000$, tolerance $10^{-7}$, and Arnoldi-Scout as the baseline. As the validation error decreases, Deepcontour obtains higher end-to-end speedup and a lower missed-eigenvalue rate. Shaded regions indicate variation across test instances.

