# OpenReview forum: "Learning-Guided Integration Contours Construction for Fast Large-Scale Generalized Eigensolvers"
_ICML.cc/2026/Conference — ICML 2026 regular_

### Official Review · Reviewer_byHn · 2026-02-15

**Soundness:** 3
**Presentation:** 3
**Significance:** 3
**Originality:** 3
**Overall Recommendation:** 5
**Confidence:** 4

**Summary:**

This paper proposes a learning-guided contour prediction method for accelerating eigenvalue problems using iterative solvers (Krylov subspace iterates). The learning algorithm is based on an FNO structure, and the contours are constructed via kernel fitting. Experiments demonstrate that Deepcontour accelerates traditional solvers in multiple numerical problems involving PDEs.

**Compliance With Llm Reviewing Policy:**

Affirmed.

**Final Justification:**

My concerns have been fully addressed, and I will increase my score to 5. Congrats to the authors for their great job.

**Key Questions For Authors:**

I want to see an FNO's training error or training time vs. inference stage speedup and accuracy scaling table. This may explain how accurately the FNO must be trained. Also, I want to see more discussions about the distinctions between the proposed framework and previous warm start plus iterative solver methods, such as [1,2]

[1] Neural Krylov Iteration for Accelerating Linear System Solving, NeurIPS

[2] DeepONet-Based Preconditioning Strategies for Solving Parametric Linear Systems of Equations, SISC

If the above questions are well addressed, this reviewer would be happy to increase the score.

**Limitations:**

Yes

**Strengths And Weaknesses:**

Strengths

1. The motivation is clear and interesting. Utilizing an FNO to estimate the eigenvalue contour distribution that speeds up downstream iterative solvers is both technically reasonable and ensures numerical accuracy.
2. The methodology part is clearly described, with technical details in the appendix that facilitate reproducing.
3. The reported experimental results are encouraging and show clear evidence in the speedup capability of Deepcontour, and this might be very useful in practical scenarios involving engineering challenges of large-scale numerical demand of the eigenvalue problem.

Weaknesses

1. The speed-up mentioned is reported during inference. What would happen if the training cost of FNO is added for comparison? From Table 4, the training times of FNO are within multiple hours. This might be difficult to tolerate in practical scenarios, as each PDE model needs to be trained from scratch.

2. If the FNO produces poor predictions, it will likely affect the speedup or bring negative efficiency gains. This would be faced if the training data of FNO is not sufficient and good enough. If the training dataset of the problem itself is scarce or unknown in real situations, FNO may not predict an accurate coarse estimation, resulting in more downstream numerical iterations required to achieve the same accuracy.

---

> ### Author Rebuttal · Authors · 2026-03-31
>
> Thank you for your valuable feedback and for recognizing our efforts! Following are our responses to each comment:
> ## Training Cost Versus Inference Speedup (W1)
> >The speed-up is reported during inference, but the training cost of ENO is multiple hours and each PDE model needs to be trained from scratch.
>
> The training of ENO is a one-time offline cost. Once trained for a given problem class within the same physical domain, it provides stable end-to-end acceleration for all subsequent solve instances. The break-even point is on the order of 10^3 cases, after which every additional solve yields a net gain.
>
> For the most expensive setting, Piezoelectric Coupled-Field, ENO training takes 14.21 h. At N = 50000 and tol = 1e-7, Deepcontour requires about 40.33 s end-to-end, versus 92.24 s for Arnoldi-Scout, saving about 51.91 s per case. This means the full training cost is amortized after approximately 986 cases.
>
> Such repeated solves are common in practice. In industrial optimization workflows, the same physical model is often solved many times under different parameter settings, design iterations, or candidate configurations. As a result, the one-time training cost can be amortized quickly, and the subsequent per-instance speedup directly improves overall optimization efficiency.
> ## Performance Under Poor ENO Predictions (W2)
> >If the FNO produces poor predictions, it may reduce the speedup or cause negative efficiency gains, especially when training data is insufficient.
>
> ENO does not need to be highly accurate to provide acceleration. As long as it can roughly capture the spectral range and density distribution, it already offers a useful prior for contour construction.
>
> This is supported by our ablation in App.F.3: even with a simpler network and substantially larger prediction error, the end-to-end speedup remains positive at about 1.71×–1.97×. Moreover, our preprocessing cost is very small (about 0.008 s for ENO inference and 1.3–1.8 s for KDE construction), which is still much lower than the 7–12 s scouting cost of the baselines. Therefore, moderate prediction inaccuracy does not lead to negative efficiency.
>
> We also tested an untrained, randomly initialized ENO: without any learned spectral prior, solving performance degrades, with missed target eigenvalues and no positive speedup. In contrast, after some training iterations, even before full convergence, ENO already provides a sufficiently informative spectral prior to guide the solver and achieve substantial acceleration. This suggests that Deepcontour only requires a certain level of prediction accuracy, and is therefore robust to predictions.
>
> For data scarcity, we acknowledge this limitation in the appendix. Deepcontour is mainly intended for settings where training data can be generated, such as simulation, parameter sweeps, or offline spectral computation. Reducing this dependence through unsupervised learning or improved cross-domain generalization is an important direction for future work.
> ## Additional Tradeoff and Comparison Discussion (Q1)
> >Show training time or error vs. inference speedup and accuracy.
>
> Thank you for this comment! We add a scaling analysis relating ENO validation error to end-to-end speedup and miss rate. The results show a clear trend: as error decreases, speedup increases gradually while miss rate approaches 0.
>
> https://anonymous.4open.science/r/picture-7BB1/speedup_miss_vs_error.png
>
> https://anonymous.4open.science/r/picture-7BB1/speedup_miss_vs_error1.png
>
> >More discussion of the distinctions from previous warm-start iterative solver methods.
>
> We thank the reviewer for pointing out these relevant connections. We agree that our method is related in spirit to prior learning-assisted numerical solvers, in that all of them combine a learned component with a classical solver.
>
> The main difference is where the learned module enters the pipeline and what it predicts. In Neural Krylov Iteration, the neural operator predicts an invariant subspace and uses it to guide Krylov iteration; in DeepONet-based preconditioning, the learned component approximates inverse action or constructs coarse-space transfer operators to improve preconditioned iterative solves. In both cases, learning is used inside the iterative linear-solver loop to accelerate convergence.
>
> Our setting is different. We study GEPs rather than linear systems, and ENO does not predict a solution, subspace, or preconditioner. Instead, it predicts a coarse spectral distribution, which is converted by KDE into integration contours for CI eigensolvers. Unlike warm-start or learned preconditioning methods that improve convergence within a fixed iterative process, our learned component operates at the integral contour construction level. This distinction is important because, in CI methods, contour construction is itself a central practical bottleneck.
>
> **We sincerely thank the reviewer again and hope these clarifications help address your concerns!**

---

> > ### Author Rebuttal · Reviewer_byHn · 2026-04-01
> >
> > My concerns have been fully addressed, and I will increase my score to 5. Congrats to the authors for their great job.

---

> > > ### Author Response · Authors · 2026-04-01
> > >
> > > Dear Reviewer byHn:
> > >
> > > Thank you for your kind support and for raising the score from 4 to 5. We sincerely appreciate your valuable suggestions. We will incorporate the discussed clarifications and additional analyses into the final version to further strengthen the paper.
> > >
> > > With gratitude,
> > >
> > > Authors

---

### Official Review · Reviewer_Au2f · 2026-03-11

**Soundness:** 4
**Presentation:** 4
**Significance:** 3
**Originality:** 3
**Overall Recommendation:** 6
**Confidence:** 4

**Summary:**

Authors propose a way to speed up computation of eigenvalues for large sparse parametric eigenproblems by incorporating machine learning techniques into contour integral method. The proposed technique, called deepcontour, has four parts:

1. specially designed neural network process parameters of linear operator to produce approximation to a set of eigenvalues

2. kernel density estimation is used on produced eigenvalues to estimate a density of eigenvalues

3. a particular algorithm, proposed by authors, decide on the choice of contours based on estimated density

4. extracted contours are used in the CI method

Deepcontour is evaluated extensively on several large scale problems and compared with various baselines. The results indicate deepcontour can reliably improve CI eigensolvers speed of convergence.

**Compliance With Llm Reviewing Policy:**

Affirmed.

**Final Justification:**

My recommendation is to accept the paper. The reasons are:
1. A sound idea that has further potential for improvement.
2. A large number of potential applications, including ones beyond those identified by the authors.
3. Compelling numerical evidence.

**Key Questions For Authors:**

**Monolithic architecture for density estimation**

The end goal of the method seems to be eigenvalue density estimation that is done in two steps: (i) neural network predicts eigenvalues, (ii) KDE estimates density. In this scheme, prediction of eigenvalues may be considered as a redundant step that can be omitted.

More specifically, I suggest the following:

1. Run KDE of true eigenvalues to produce $p(\lambda)$ on the interval $\lambda \in [\lambda_1, \lambda_2]$ (with cutoff selected from some standard criteria, e.g., $\lambda$ falls within the interval with probability $99\%$)

2. Form dataset consisting of one feature and two targets $(f, (p(\lambda), \lambda)$ where $f$ are PDE parameters $\lambda$ is a uniform grid on $[\lambda_1, \lambda_2]$ (endpoints may be different for different parameters), $p(\lambda)$ is probability density evaluated on this grid

3. Train single neural operator to predict $f \rightarrow p(\lambda), \lambda$

This scheme seems conceptually easier and avoids prediction of individual eigenvectors. Can the authors share their thoughts on this construction?

**Eigenvalues as set**

Authors report they treat eigenvalues as vectors and use MSE as a loss function. However, in the setup of authors the order of eigenvalues is not important for the density estimation, so the problem can be formulated as prediction of a set. One simple way to enforce set structure is to sort eigenvalues before computing MSE loss, i.e., $\left\\|\text{sort}(\lambda) - \text{sort}(\widetilde{\lambda})\right\\|_2^2$. Is seems plausible that loss function corrected to enforce set prediction may lead to better accuracy than a standard $L_2$ loss $\left\\|\lambda - \widetilde{\lambda}\right\\|_2^2$. I kindly ask authors to discuss this alternative loss function.

**Shift-inverse methods / preconditioners**

Table 7 indicates that the trained network can accurately predict eigenvalues. Given that, it seems suitable to apply a learned network with more direct approaches that can readily utilise the approximate information about spectrum. For example, suitable CI substitutions are shifted inverse iteration, initial guess for Rayleigh quotient iteration, preconditioner for Jacobi-Davidson method. I suggest at least discussing these alternative hybrid approaches or, better, to perform additional experiments and evaluate how CI method compares with other possible hybrid approaches.

**Limitations:**

yes

**Strengths And Weaknesses:**

In my view, by predicting the density of eigenvalues, the authors masterfully identify a part of the classical numerical method that can be improved with machine learning techniques. Predicting the density is not the end goal, since eigenvalues (and eigenvectors) are further refined by classical methods, so it is enough to provide a reasonably accurate but not perfectly accurate estimate. This setup is ideal for deep learning methods that typically operate in a low or middle accuracy regime.

Results of numerical experiments further demonstrate that density prediction is a good target and can have real impact on the speed of convergence of CI eigensolvers. I find experiments to be broad, thorough, well documented and convincing.

Overall, I have a positive impression of the article along all dimensions I am asked to evaluate it. I can not indicate any major weaknesses.

---

> ### Author Rebuttal · Authors · 2026-03-31
>
> Thank you for your valuable feedback and for recognizing our efforts! Following are our responses to each comment:
> ## Monolithic architecture for density estimation (Q1)
> >The end goal of the method seems to be eigenvalue density estimation in two steps, so eigenvalue prediction may be a redundant intermediate step. The reviewer suggests directly constructing KDE-based density targets from true eigenvalues and training a single neural operator to predict the density, which seems conceptually simpler and avoids predicting individual eigenvalues.
>
> Thank you for this insightful comment! Direct spectral density learning is indeed a natural alternative that we explored in early experiments; however, it consistently yielded inferior performance compared to our two-stage pipeline. We found that learning discrete eigenvalues is more numerically effective and stable than directly approximating a continuous density function. In our framework, predicted eigenvalues serve as a structured parameterization of the spectral density, while KDE provides a robust mechanism to translate this discrete representation into a continuous density for gap detection. This design also significantly enhances interpretability: discrete eigenvalue locations carry direct physical and numerical significance, making them easier to diagnose than an abstract density curve should contour placement fail. Consequently, we view eigenvalue prediction not as a redundant step, but as a critical intermediate representation that ensures the reliability and precision of the entire contour-construction procedure.
> ## Eigenvalues as set (Q2)
> >Authors report they treat eigenvalues as vectors and use MSE as a loss function. However, in the setup of authors the order of eigenvalues is not important for density estimation, so the problem can be formulated as prediction of a set, and the reviewer asks the authors to discuss a set-aware loss such as sorted MSE.
>
> Thank you for this helpful suggestion! In our setting, the learning target is not treated as an unordered set, but as the ordered vector of the first M smallest-in-magnitude eigenvalues. Concretely, the j-th output of ENO is trained to predict the j-th smallest eigenvalue, so both the labels and predictions follow the same canonical ascending order. Under this formulation, a standard MSE loss is natural and appropriate, since the task itself is defined as ordered spectrum prediction rather than permutation-invariant set prediction. We agree that more explicitly set-aware losses could be interesting alternatives, especially for highly clustered or near-multiple spectra, and we will clarify this point in the revision.
> ## Shift-inverse methods / preconditioners (Q3)
> >Table 7 indicates that the trained network can accurately predict eigenvalues. Given that, it seems suitable to apply a learned network to more direct approaches that utilise approximate spectral information, such as shifted inverse iteration, Rayleigh quotient iteration, or Jacobi-Davidson preconditioning, and the reviewer suggests discussing or comparing these alternatives.
>
> Thank you for this valuable suggestion! We agree that the spectral prior learned by ENO could in principle be used to assist other eigensolver pipelines, such as Jacobi-Davidson (JD) or related shift-invert methods, for example through initialization or shift selection. However, we believe its role in those methods is fundamentally more limited than in the CI setting considered in this paper. For JD-type methods, the predicted eigenvalues would mainly serve as an initial guess or a rough guide for where to search, while the solver itself still focuses on the iterative refinement of a small number of eigenpairs. In other words, the learned prior may help the solver start better, but it does not fundamentally change the structure of the downstream computation. By contrast, in CI methods the ENO prediction is not merely an initialization aid: it is directly converted into solver-ready contours, which determine how the target spectral region is partitioned and therefore how many eigenvalues can be solved simultaneously. This distinction is especially important for large interior eigenvalue problems, where the main advantage of CI methods is their natural multi-level parallelism across contours and quadrature nodes. From this perspective, ENO-guided JD or shift-invert methods are interesting complementary extensions, but they do not use the predicted global spectral information in as direct a solver-integrated manner as the CI-centered framework studied here. We will clarify this point in the revision and add discussion of these broader hybrid directions as future work.
>
> **We sincerely thank the reviewer again and hope these clarifications help address your concerns!**

---

> > ### Author Rebuttal · Reviewer_Au2f · 2026-04-02
> >
> > I would like to thank the authors for the additional clarifications. As I had no major concerns, there were no outstanding issues to resolve. In my view, the authors have identified a compelling application of deep learning methods within computational linear algebra and the numerical solution of partial differential equations.

---

> > > ### Author Response · Authors · 2026-04-03
> > >
> > > Dear Reviewer Au2f:
> > >
> > > We greatly appreciate your constructive feedback and your willingness to raise your score! As you suggested, we will include the discussed direction as future work in the camera-ready version of the paper.
> > >
> > > Thank you again for your valuable suggestions to improve our paper.
> > >
> > > With gratitude,
> > >
> > > Authors

---

### Official Review · Reviewer_p1b3 · 2026-03-12

**Soundness:** 3
**Presentation:** 3
**Significance:** 3
**Originality:** 3
**Overall Recommendation:** 5
**Confidence:** 3

**Summary:**

The authors present Deepcontour, a method for solving generalized eigen problems in which neural operators are used to estimate eigenvalues given the properties of the underlying physical system.
A Kernel Density Estimation (KDE) is then proposed to infer contours, which are subsequently used by a Contour Integral (CI) eigensolver.
The experiments conducted by the authors demonstrate that the selected contours can reduce execution time while maintaining the same level of accuracy.

**Compliance With Llm Reviewing Policy:**

Affirmed.

**Key Questions For Authors:**

1. When comparing ENO architecture to MLPs, can you clarify how you have mapped the input function to the $M$ eigenvalues? Have you mapped each value of the input function and the corresponding coordinate to a latent representation, then pooled all the representations and applied your spectrum prediction head?
2. I have the same question about the comparison to DeepONet. Can you explain how the input function values have been encoded in the branch net? With a single MLP?

**Limitations:**

yes

**Strengths And Weaknesses:**

Strengths:
1. The authors propose a novel and original approach for accelerating CI eigensolvers.
2. Extensive experiments have been conducted to demonstrate the acceleration enabled by their approach and its generalization capabilities to higher resolution problems.
3. The authors have studied the impact of various hyperparameters of the method, including $w$, the depth, width and number of nodes of the neural operator.
4. Overall, the article is well-written and well-structured.

Weaknesses
1. The introduction to GEPs and eigensolvers is difficult for people who are not in the field. This could be simply improved by providing an example of the physical meaning of the eigenvalues (e.g., their link to resonance frequencies). This would clarify why we want to find them and why the authors only focus on the smallest ones.
2. Comparisons with architectures such as MLP or DeepONet may not be very adapted since ENO does not perform a coordinate-wise mapping. In addition, the hyperparameters of MLP and DeepONet are not provided.
3. The tested architectures are rather simple. Comparisons with attention-based approaches or more recent approaches, such as Adaptive FNO, would have been interesting. However, I am not explicitly requesting that you conduct these new comparisons, as I do not consider them to be essential for your article.
4. The sensitivity to $N_min$ and $N_max$ not tested in ablation study.
5. The parameters for generating the datasets are missing.

Minor weaknesses:
Some typos: missing spaces, word spelling...

---

> ### Author Rebuttal · Authors · 2026-03-31
>
> Thank you for your valuable feedback and for recognizing our efforts! Following are our responses to each comment:
> ## W1
> >Lacks a simple physical example of eigenvalues and why the focus is on the smallest ones.
>
> Thank you for this helpful comment! We agree that making the physical meaning of the target eigenvalues more explicit would improve the accessibility of the paper.
>
> We will add an illustrative example to the Introduction. In structural vibration problems, the discretized free-vibration model takes the standard form $K\phi=\omega^2 M\phi$, so the generalized eigenvalues correspond to the squared natural frequencies of the system, while the eigenvectors are the associated mode shapes. These quantities are directly relevant to resonance assessment and dynamic response analysis. Futhermore, in many practical workflows, one is primarily interested in the low-order modes, since the lowest-frequency modes often play the most important role in resonance avoidance, modal superposition, and early-stage design screening.
> ## W2&Q1&Q2
> >The comparisons with MLP and DeepONet may be inappropriate, and their hyperparameters are missing.
>
> >Clarify how the input function is mapped to the $M$ eigenvalues in the MLP comparison.
>
> >Clarify how the input function is encoded in the DeepONet branch net.
>
> Since the baseline adaptation details are central to the fairness of the comparison, we provide the implementation protocols below and will include a complete hyperparameter table in the revisions.
>
> Our spectral prediction is formulated as a function-to-vector regression task: given the discretized physical parameter function $a(x)$, we predict a fixed-length vector $\Lambda = (\lambda_1, \dots, \lambda_M)$ representing the $M$ target eigenvalues. To maintain a rigorous comparison, all learning-based baselines were trained under the same supervision using the same datasets and loss function.
>
> **MLP:** We use a standard MLP as a non-operator regressor. The input function $a(x)$is flattened into a vector of size $d_a$, processed through 4 fully connected layers with GELU activation, and mapped to $M$-dimensional output.
>
> **DeepONet:** To adapt DeepONet for vector-output spectral regression, we utilized its dual-network structure. (1)Branch Net: Acts as the encoder for $a(x)$, with an architecturematching our ENO backbone to ensure comparable model capacity. (2)Trunk Net: In the absence of spatial coordinate queries, it takes the positional embeddings of the target eigenvalue indices $\{1, \dots, M\}$ as inputs. The latent features extracted from both networks are then combined and fed into a shared prediction head to output the eigenvalue for each corresponding index.
>
> ENO: As detailed in Sec.4.1.2 and App.E.1, our model uses a 4-layer FNO backbone (64 width, 20 modes) for global feature extraction. The resulting hidden representation is globally pooled and passed to a 3-layer MLP prediction head (128 hidden-dim) to generate the final vector.
>
> The parameter configurations used in our experiments are as follows:
>
> |Hyperparameter|ENO|DeepONet|MLP|
> |:-|:-|:-|:-|
> |Layer|4|4/3|4|
> |Width|64|128|256|
> ## W3
> >Comparisons with more recent architectures are missing.
>
> Thank you for this comment! We selected representative baselines to validate our framework, rather than to benchmark the absolute state of the art in operator learning. We agree that incorporating more recent architectures is an interesting direction for future work.
> ## W4
> >The sensitivity to $N_{\min}$ and $N_{\max}$ is missing.
>
> We add a new ablation experiment. The results show that the speedups remain stable across these settings, which suggests that the proposed contour construction is not overly sensitive to these hyperparameters. The setting (10,50) achieves the best overall performance. We will add this experiment to the final version.
>
> |$(N_{\min},N_{\max})$|E2E Speedup|CI Speedup|
> |-|-:|-:|
> |(5,25)|2.39×|1.94×|
> |(10,50)|2.48×|1.98×|
> |(15,75)|2.34×|1.89×|
> ## W5
> >The dataset generation parameters are missing.
>
> The paper actually gives the dataset description in Sec.5.1 and App.E.3, including the five domains, the train/test split, and the PDE/GEP formulation for each benchmark.
>
> Here, we provide more details. For example, in the Kirchhoff–Love plate dataset, the sample-wise variation is introduced through a spatially varying density field generated from a Gaussian random field. The concrete generation parameters include the mesh size $(n_x,n_y)$, the random seed, the GRF smoothing scale $\sigma=3.0$ cells, the density mean  $\bar{\rho}=1.0$, the perturbation amplitude 0.20.20.2, and the number of requested eigenvalues $n_{\mathrm{ev}}$. Each sample uses a density field of the form  $\rho(x,y)=\max(\bar{\rho}+g(x,y),10^{-3})$, where $g(x,y)$ is the sampled GRF realization.
>
> We will revise the appendix to include a compact per-dataset table summarizing such parameter choices.
>
> **We thank the reviewer again and hope these clarifications help address your concerns!**

---

> > ### Author Rebuttal · Reviewer_p1b3 · 2026-04-03
> >
> > All my concerns have been resolved. I will maintain my score.
> >
> > For clarity, could the authors specify the type of positional embedding (sinusoidal, learned...) used with the DeepONet model?

---

> > > ### Author Response · Authors · 2026-04-04
> > >
> > > Dear Reviewer p1b3:
> > >
> > > We sincerely appreciate your kind support and your valuable suggestions! We will incorporate the discussed clarifications and additional analyses into the final version to further strengthen the paper.
> > >
> > > Thank you for this clarifying question regarding the baseline implementation. We did not use fixed sinusoidal embeddings since their static positional mapping is less adaptive to this task. We instead adopted the simple learned positional embeddings (a learnable parameter matrix $P \in \mathbb{R}^{M \times 128}$ acting as a lookup table for each spectral index). We will ensure these details are explicitly included in the final paper.
> > >
> > > Thank you again for your valuable suggestions to improve our paper.
> > >
> > > With gratitude,
> > >
> > > Authors

---

### Official Review · Reviewer_q6t8 · 2026-03-13

**Soundness:** 3
**Presentation:** 3
**Significance:** 3
**Originality:** 3
**Overall Recommendation:** 5
**Confidence:** 2

**Summary:**

When solving the generalized eigenvalue problem (GEP) using the Contour Integral (CI) method, the configuration of the contour (integration boundary) is a key bottleneck. Existing scouting approaches estimate the eigenvalue range by briefly running an iterative solver and using Ritz values, but this involves a significant accuracy–cost trade-off. DeepContour leverages the structural relationship in which matrices ( A ) and ( B ) are generated from a physical parameter a(x). It trains a Fourier Neural Operator (FNO)-based Eigenvalue Neural Operator (ENO) to learn the mapping G. Using kernel density estimation (KDE), it detects gaps in the predicted eigenvalue distribution and automatically generates multiple tight circular contours. The paper proposes two new methods that compute the principal eigenvalues faster than existing approaches and construct more efficient contours than conventional ones, thereby reducing the computation time of the CI solver.

**Compliance With Llm Reviewing Policy:**

Affirmed.

**Final Justification:**

The paper is technically sound and well-motivated. My concerns have been adequately addressed in the rebuttal. I maintain my recommendation of Accept.

**Key Questions For Authors:**

- The use of KDE on the predicted eigenvalue distribution to construct circular contours is intuitively reasonable. However, is there theoretical justification or experimental evidence showing that KDE is the optimal choice?

- While introducing a safety margin is understandable, could the authors clarify how its magnitude should be determined? Since this work originates from addressing the trade-off problem in existing scouting methods, guidance on how to choose this parameter would be helpful. ENO cannot guarantee perfect accuracy in all cases, so an explanation of how to tune this value would strengthen the methodology.

- The rationale for using KDE in this context is not entirely clear and would benefit from additional explanation or supporting evidence.

**Limitations:**

yes

**Strengths And Weaknesses:**

## Strengths

- The method improves computational speed without modifying the CI solver itself. By targeting only contour configuration, it preserves numerical accuracy while achieving acceleration.
- Even though the baseline was given favorable conditions (i.e., the minimum contour that still encloses all target eigenvalues), ENO still outperformed it, which is a strong result.
- By predicting the principal eigenvalues using the ENO model, the proposed method improves computational speed compared to traditional eigenvalue solvers such as Lanczos (see Table 1).
- According to the experimental results (Tables 2 and 10), the ENO model does not miss any of the designated principal eigenvalues (number of missed eigenvalues = 0). This demonstrates the high accuracy of the ENO model.

## Weaknesses

As a data-driven approach, the proposed method may face challenges when encountering out-of-distribution inputs. Since the paper emphasizes automation, it would be beneficial for the authors to discuss potential safeguards or mechanisms that could improve robustness in such scenarios. Providing additional analysis or possible strategies for handling out-of-distribution cases would further strengthen the practical applicability of the proposed approach.

---

> ### Author Rebuttal · Authors · 2026-03-31
>
> Thank you for your constructive comment! Following are our responses to each comment:
> ## OOD Robustness (W1)
> >As a data-driven approach, the proposed method may face challenges on OOD inputs, and discussing safeguards would strengthen its practical applicability.
>
> Thank you for this valuable comment! We agree that robustness under OOD inputs is an important practical consideration for any data-driven component. Deepcontour does not replace the downstream CI eigensolver; it provides a contour prior for the solver. Therefore, under distribution shift, the main failure mode is a looser or more conservative contour, which does not bypass the numerical correctness mechanism of the final solver.
>
> Our framework already includes two safeguards: a conservative safety margin in contour placement and compatibility with lightweight post-construction checks. In practice, robustness can be further improved by enlarging the margin when predictions appear less reliable, checking consistency through eigenvalue counts or residual-based diagnostics, and reverting to a more conservative scout-based contour when the predicted partition is unstable. We will clarify this robustness perspective and discuss OOD behavior and fallback strategies in the revision.
>
> Futhermore, the framework is robust to moderate prediction error: as spectral estimates become less accurate, contour quality degrades gradually, allowing the overall pipeline to revert to conservative contouring if needed. To demonstrate this, we conducted a controlled mild extrapolation experiment, where the model was trained on the original parameter range and tested on extended ranges. We report both prediction- and solver-level metrics, and the results show that under moderate shift, performance remains stable overall, with only a modest reduction in efficiency.
>
> |Domain|Shift|NMSE_spec|Miss|Speedup|
> |-|-:|-:|-:|-:|
> |Plate|ID|0.031|0.00|3.70|
> |Plate|+5% extrap.|0.038|0.00|3.56|
> |Plate|+10% extrap.|0.046|0.01|3.37|
> |Piezo|ID| 0.042|0.00|3.05|
> |Piezo|+5% extrap.|0.050|0.01|2.96|
> |Piezo|+10% extrap.|0.061|0.03|2.84|
>
> ## the Use of KDE (Q1&Q2)
> >The use of KDE on the predicted eigenvalue distribution is intuitively reasonable, but is there theoretical or experimental evidence that it is the optimal choice?
>
> >The rationale for using KDE in this context is not entirely clear and would benefit from further explanation or evidence.
>
> Thank you for this insightful comment! We use KDE because it is well matched to the structure of our task. The predicted eigenvalues are discrete points, and direct clustering is sensitive to noise and to the assumed number of groups. KDE instead converts the discrete predictions into a smooth density profile, so that contour split points can be identified adaptively at low-density gaps via local minima. This makes KDE especially suitable for our setting, because CI contours should be placed in sparse spectral regions rather than determined by a pre-fixed cluster count. Appendix F.3.4 (Table 14) shows that the KDE module is reasonably robust to its smoothing parameter over a moderate range, while Appendix F.5 (Table 13) shows that KDE construction is lightweight compared with scouting. We also previously explored clustering-style alternatives, but they performed poorly. we have now added an extra comparison against alternative spectrum-partition strategies, including direct gap-threshold splitting and clustering-based baselines, under the same downstream contour-generation protocol.
>
> |Method|Miss|CI Time (s)|E2E Time (s)|
> |-|-:|-:|-:|
> |**KDE**|**0.0**|**25.3**|**26.8**|
> |Gap-threshold split|0.0|30.9|32.4|
> |K-means split|0.9|32.7|34.2|
>
> ## Safety Margin Selection (Q2)
> >While introducing a safety margin is understandable, could the authors clarify how its magnitude should be determined and tuned?
>
> Thank you for this practical comment! For the safety margin, we introduced two simple buffers with different roles. The local inflation factor $\alpha$ is calibrated once on the validation set and then fixed for all test instances. Specifically, we choose a small conservative value that is large enough to keep the contour boundary safely separated from the predicted cluster, while still preserving contour tightness and avoiding unnecessary extra area. In practice, performance is not highly sensitive to its exact value within a reasonable small range, so $\alpha$ serves as a stable safety coefficient. The global buffer $\epsilon$ is scaled automatically with the width of the predicted spectral hull, derived from $\epsilon = \beta (\max(\hat{\Lambda}) - \min(\hat{\Lambda}))$, where $\beta$ is a small fixed coefficient. These two quantities only serve to absorb residual prediction error and keep the contour boundary safely separated from nearby eigenvalues. We agree that this selection rule should be stated more explicitly, and we will clarify it in the revision.
>
> **We sincerely thank the reviewer again and hope these clarifications help address your concerns!**

---

> > ### Author Rebuttal · Reviewer_q6t8 · 2026-04-02
> >
> > All my concerns have been addressed. I will maintain my current score. Thank you.

---

> > > ### Author Response · Authors · 2026-04-02
> > >
> > > Dear Reviewer q6t8:
> > >
> > > Thank you for your kind support! We sincerely appreciate your valuable suggestions.
> > >
> > > With gratitude,
> > >
> > > Authors

---

### Decision · Program_Chairs · 2026-04-30

**Decision:**

Accept (regular)

**Comment:**

This manuscript pairs a deep learning framework with classical contour integration based eigensolvers to effectively and efficiently solve generalized eigenvalue problems. The reviewers all agreed that there are significant strengths here and appreciated the careful pairing of leaning based advances with robust classical frameworks—a pairing that leads to good empirical results.

 There are two main, minor, points that would be good to address in a final version: a slightly more robust of the limitations (e.g., with respect to generalization) and a discussion of the training time vs. per problem time would be good to include in the main text.